# LARGE LEARNING RATE MATTERS FOR NON-CONVEX OPTIMIZATION

## ABSTRACT

When training neural networks, it has been widely observed that a large step size is essential in stochastic gradient descent (SGD) for obtaining superior models. However, the effect of large step sizes on the success of SGD is not well understood theoretically. Several previous works have attributed this success to the stochastic noise present in SGD. However, we show through a novel set of experiments that the stochastic noise is not sufficient to explain good non-convex training, and that instead the effect of a large learning rate itself is essential for obtaining best performance. We demonstrate the same effects also in the noise-less case, i.e. for full-batch GD. We formally prove that GD with large step size—on certain non-convex function classes—follows a different trajectory than GD with a small step size, which can lead to convergence to a global minimum instead of a local one. Finally, we also demonstrate the difference in trajectories for small and large learning rates for real neural networks, again observing that large learning rates allow escaping from a local minimum, confirming this behavior is indeed relevant in practice.

## 1 INTRODUCTION

While using variants of gradient descent (GD), namely stochastic gradient descent (SGD), has become standard for optimizing neural networks, the reason behind their success and the effect of various hyperparameters is not yet fully understood. One example is the practical observation that using a large learning rate in the initial phase of training is necessary for obtaining well performing models (Li et al., 2019). Though this behavior has been widely observed in practice, it is not fully captured by existing theoretical frameworks.

Recent investigations of SGD's success (Kleinberg et al., 2018; Pesme et al., 2021) have focused on understanding the implicit bias induced by the stochasticity. Note that the effective variance of the trajectory due to the stochasticity of the gradient is moderated by the learning rate (see Appendix F for more intuition). Therefore, using a larger learning rate amplifies the stochasticity and the implicit bias induced by it which can provide a possible explanation for the need for larger learning rates. We show that this explanation is incomplete by demonstrating cases where using stochasticity with arbitrary magnitude but with a small learning rate, can not guarantee convergence to global minimum whereas using a large learning rate can. Furthermore, we provide a practical method to increase stochasticity without changing the learning rate when training neural networks and observe that increased stochasticity can not replace the effects of large learning rates. Therefore, it is important to study how a larger learning rate affects the trajectory beyond increasing the stochasticity.

To that end, in this work we show that randomly initialized gradient descent with a high learning rate provably escapes local minima and converges to the global minimum over of a class of non-convex functions. In contrast, when using a small learning rate, GD over these functions can converge to a local minimum instead. We note that for brevity, we focus our results on the full-batch GD.

We further show the positive effect of using a high learning rate to increase the chance of completely avoiding undesirable regions of the landscape such as a local minimum. Note that this behavior does not happen when using the continuous version of GD, i.e. gradient flow which corresponds to using infinitesimal step sizes. The difference remains even after adding the implicit regularization term identified in (Smith et al., 2021) in order to bring trajectories of gradient flow and gradient descent closer.

We would like to note that throughout the paper, we sometimes misuse the terms "global" and "local" minimum to refer to desirable and undesirable minima respectively. For example when discussing generalization, a desirable minimum might not have the lowest objective value but enjoy properties such as flatness.

Finally, to show the relevance of our theoretical results in practice, we demonstrate similar effects can happen in neural network training by showing evidence of an escape from local minimum when applying GD with a high learning rate on a commonly used neural network architecture. Our observations signify the importance of considering the effects of high learning rates for understanding the success of GD.

Overall, our contributions can be summarized as follows:

- Demonstrating the exclusive effects of large learning rates even in the stochastic setting both in theory and in practice, showing that they can not be reproduced by increasing stochasticity and establishing the importance of analyzing them.
- Capturing the distinct trajectories of large learning rate GD and small learning rate GD in theory on a class of functions, demonstrating the empowering effect of large learning rate to escape from local minima.
- Providing experimental evidence showing that gradient descent escapes from local minima in neural network training when using a large learning rate, establishing the relevance of our theoretical results in practice.

## 2 RELATED WORK

Extensive literature exists on studying the effect of stochastic noise on the convergence of GD. Several works have focused on the smoothing effect of injected noise (Chaudhari et al., 2017; Kleinberg et al., 2018; Orvieto et al., 2022; Wang et al., 2021a). In (Vardhan & Stich, 2022) it has been shown that by perturbing the parameters at every step (called perturbed GD) it is possible to converge to the minimum of a function $f$ while receiving gradients of $f + g$, assuming certain bounds on $g$. Other works use different models for the stochastic noise in SGD and use it to obtain convergence bounds or to show SGD prefers certain type (usually flat) of minima (Wu et al., 2018; Xie et al., 2021). In order to better understand the effect of various hyperparameters on convergence, Jastrzebski et al. (2019); Jastrzbski et al. (2018) show the learning rate (and its ratio to batch size) plays an important role in determining the minima found by SGD. In (Pesme et al., 2021) it was shown that SGD has an implicit bias in comparison with gradient flow and its magnitude depends on the learning rate. While this shows one benefit of using large learning rates, in this work, we provide evidence that the effect of learning rate on optimization goes beyond controlling the amount of induced stochastic noise.

Prior work also experimentally establish existence of different phases during training of a neural network. Cohen et al. (2021) show that initially Hessian eigenvalues tend to grow until reaching the convergence threshold for the used learning rate, a state they call "Edge of Stability". This growth is also reported in (Lewkowycz et al., 2020) for the maximum eigenvalue of the Neural Tangent Kernel (Jacot et al., 2018) where it has also been observed that this value decreases later in training, leading to convergence. Recent works have also investigated GD's behavior at the edge of stability for some settings (Arora et al., 2022) obtaining insights such as its effect on balancing norms of the layers of a two layer ReLU network (Chen & Bruna, 2022). In our results, GD is above the conventional stability threshold while it is escaping from a local minimum but returns to stability once the escape is finished.

In (Elkabetz & Cohen, 2021) it is conjectured that gradient descent and gradient flow have close trajectories for neural networks. However, the aforementioned observations suggest that gradient descent with a large learning rate visits a different set of points in the landscape than GD with a small learning rate. Therefore, this conjecture might not hold for general networks. The difference in trajectory is also supported by the practical observation that a large learning rate leads to a better model (Li et al., 2019).

To bridge this gap and by comparing gradient flow and gradient descent trajectories, Barrett & Dherin (2021) identify an implicit regularization term on gradient norm induced by using discrete steps. Still, this term is not enough to remove a local minimum from the landscape. Other implicit regularization terms specific to various problems have also been proposed in the literature (Ma et al., 2020; Razin & Cohen, 2020; Wang et al., 2021b). In this paper, we provide experimental evidence

and showcase the benefits of using large step sizes that are unlikely to be representable through a regularization term, suggesting that considering discrete steps might be necessary to understand the success of GD.

The type of obstacles encountered during optimization of a neural network is a long-standing question in the literature. Lee et al. (2016) show that gradient descent with random initialization almost surely avoids saddle points. However it is still unclear whether local minima are encountered during training. In (Goodfellow & Vinyals, 2015) it was observed that the loss decreases monotonically over the line between the initialization and the final convergence points. However, it was later shown that this observation does not hold when using larger learning rates (Lucas et al., 2021). Swirszcz et al. (2017) also show that it is possible to create datasets which lead to a landscape containing local minima. Furthermore, better visualization of the landscape shows non-convexities can be observed on some loss functions (Li et al., 2018). For the concrete case of two layer ReLU networks, Safran & Shamir (2018) show gradient descent converges to local minima quite often without the help of over-parameterization. Also, it was shown that in the over-parameterized setting, the network is not locally convex around any differentiable global minimum and one-point strong convexity only holds in most but not all directions (Safran et al., 2021). These observations show the importance of understanding the mechanisms of escaping local minima. We also use these observations to make assumptions that are practically justifiable.

There also exists a body of work on which properties of a minimum leads to better generalization (Dinh et al., 2017; Dziugaite & Roy, 2017; Keskar et al., 2017; Tsuzuku et al., 2020). In this work, our goal is to show the ability of gradient descent to avoid certain minima when using a high learning rate. However, the argument about whether these minima offer better or worse generalization is outside the scope of this work.

## 3 MAIN RESULTS

**Theoretical Proof of Escaping From Local Minima with a Large Learning Rate**    The need for a large learning rate in practice is commonly explained based on the intuition of escaping certain local minima. However, a theoretical setting where GD escapes from a local minimum and converges to a global minimum is lacking. Such settings are necessary both for understanding success of GD and for analyzing the effectiveness of other optimizers. In this work, we introduce a class of functions where such behavior can be observed from GD. This is stated in Theorem 1 which we describe here informally and leave the formal version to Section 4.1.

**Theorem 1** (Informal). *There exists a class of functions having at least two minima $\mathbf{x}^{\dagger}$ and $\mathbf{x}_{\star}$ where GD initialized on a random point, converges to $\mathbf{x}_{\star}$ with a large learning rate almost surely but might converge to $\mathbf{x}^{\dagger}$ with a small learning rate.*

**Theoretical Analysis of Avoiding Local Minima**    As an alternative to escaping from minima, we note that due to discrete steps in GD, it may not visit any point in an arbitrary but small part of the landscape $X$, such as a local minimum. However, note that there may still exist a set of starting points for which GD iterates reach a point in $X$. Therefore, assuming the starting point is chosen randomly, not visiting any point in $X$ is a probabilistic event. In this work, we provide a lower bound for the probability of this event in Theorem 2 which we state here informally and postpone the formal statement to Section 4.2.

**Theorem 2** (Informal). *For any arbitrary part (subset) of the landscape $X$ sufficiently far from the global minimum, let $E_X$ be the probabilistic event that GD, when initialized randomly from a large enough set, will not iterate over any point in $X$. Then under certain assumptions on the landscape, $\Pr[E_X]$ can be lower bounded where the bound depends monotonically increasing on the learning rate and inversely on the size of $X$ (as measured by Lebesgue measure). In particular, if $X$ is finite, this probability is 1.*

The dependence of the lower bound on the learning rate is intuitive as a larger learning rate allows larger steps and makes it less probable (but not impossible) to visit a small part of the landscape as illustrated in Figure 1. We note that avoiding a region is inherently different from escaping from it. In particular, as can be seen in the example, this region can be almost completely flat. In this case, once that GD reaches a point in this region, it will instantly converge. Furthermore, the region can even contain points where the function can not be differentiated. Therefore, such effect can not be compensated for by adding previously identified implicit regularization terms such as the one in

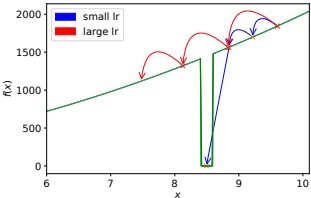 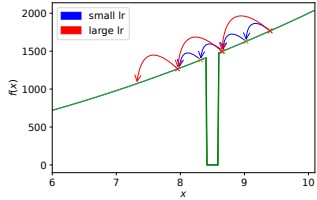 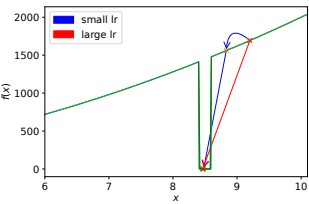

(a) GD escapes with large LR.    (b) GD escapes with both LRs.    (c) GD does not escape.

Figure 1: Success of GD to avoid a region based on the magnitude of learning rate when initialized from different points. While various cases are possible, it is more likely to avoid the minimum with a higher learning rate.

(Smith et al., 2021). This suggests other methods are needed to bridge the gap between gradient flow and GD. We demonstrate both effects of escaping and avoiding local minima on an example function in Appendix H.

**Importance of Large Learning Rate Despite the Effects of Stochastic Noise**   One possible explanation for the need of large learning rate is the magnification of the effect of stochastic noise (see Appendix F for further intuition), facilitating escaping from local minima. While this explanation can not explain the success of full-batch GD, it is more common to use SGD in practice. Therefore, this explanation makes it questionable whether it is necessary to understand direct effects of learning rate on the trajectory or is it enough to only consider the stochastic noise.

In this work, we show that the effects of using a large learning rate goes beyond magnifying stochastic noise. To that end, we first provide an example in Section 4.3 where escaping from a local minimum and converging to the global minimum can only be achieved with a large learning rate even in presence of stochastic noise. Furthermore, we demonstrate this result in practice in Section 5.1 by decoupling the effect of stochastic noise on the trajectory and the magnitude of the learning rate when training neural networks. Our experiment results show that the effects of the large learning rate remain crucial for converging to the correct minimum even in presence of (magnified) stochastic noise.

**Demonstrating Effects of Large Learning Rate in Neural Networks**   While escaping local minima is an intuitive explanation, an escape is not clearly observed while training a neural network using GD with a large learning rate even though it converges to a different minimum. In particular, it seems GD automatically avoids cases where it gets close to a local minima and then escapes from it. This makes it hard to verify the relevance of escaping behavior for neural network landscape. We do so in Section 5.2 by deliberately finding a point close to a minimum that GD would converge to with a small learning rate. In contrast, when applying GD with a large learning rate from this point, we can clearly observe an escape both in the trajectory and in the value of the loss.

## 4 THEORETICAL ANALYSIS

We now state our results more formally. For our theoretical analysis, we focus on optimizing the minimization problem

$$f_\star := \min_{\mathbf{x} \in \mathbb{R}^d} f(\mathbf{x})$$

using (full-batch) gradient descent with random initialization. For completeness, we provide a pseudo code in the Appendix A, Algorithm 1.

As is done widely in the literature, we assume smoothness (as defined in Definition 1) over regions of the landscape to ensure the gradient does not change too sharply.

**Definition 1** (*L*-smoothness). *A function $f \colon \mathbb{R}^d \to \mathbb{R}$ is L-smooth if it is differentiable and there exists a constant $L > 0$ such that:*

$$\|\nabla f(\mathbf{x}) - \nabla f(\mathbf{y})\| \le L \|\mathbf{x} - \mathbf{y}\|, \qquad \forall \mathbf{x}, \mathbf{y} \in \mathbb{R}^d. \tag{1}$$

Similarly, we need to ensure sharpness of certain regions, in particular around a local minima, to obtain our results. Therefore, to ensure a lower bound for sharpness in our analysis, we use one-point strong convexity assumption on these regions as defined in the following definition which also commonly appears in the literature:

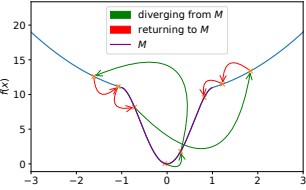
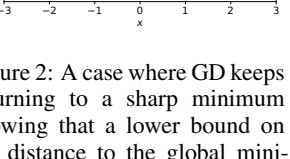

Figure 2: A case where GD keeps returning to a sharp minimum showing that a lower bound on the distance to the global minimum might be necessary to show it can be avoided.

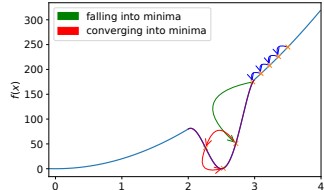
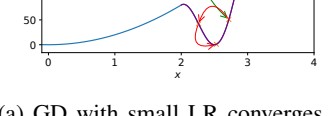

(a) GD with small LR converges to the minimum.

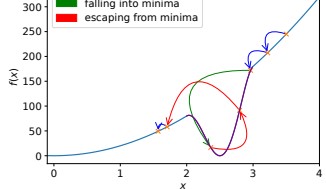

(b) GD with large LR escapes the minimum.

Figure 3: Different behaviors of GD based on the magnitude of learning rate in escaping or converging a sharp minima. GD with a high enough learning rate always escapes the minimum.

**Definition 2** ($\mu$-one-point-strongly-convex (OPSC) with respect to $\mathbf{x}_\star$ over $M$). *A function $f \colon \mathbb{R}^d \to \mathbb{R}$ is one-point strongly convex with respect to $\mathbf{x}_\star$ if it is differentiable and there exists a constant $\mu > 0$ such that:*

$$\langle \nabla f(\mathbf{x}), \mathbf{x} - \mathbf{x}_\star \rangle \geq \mu \|\mathbf{x} - \mathbf{x}_\star\|^2, \qquad \forall \mathbf{x} \in M . \tag{2}$$

Assuming OPSC property is common in the literature. When this assumption is applied over the whole landscape, it has been shown to guarantee convergence to $\mathbf{x}_\star$ (Kleinberg et al., 2018; Lee et al., 2016; Safran et al., 2021). However, in this work we only make this assumption hold on a limited part of the landscape, namely regions around a local minima. Furthermore, we use this assumption to ensure sharpness which we show can result in escaping from the regions where this assumption holds rather than converging to them. Note that recent works have verified both theoretically and empirically that landscapes of neural networks satisfy this property to some extent (Kleinberg et al., 2018; Safran et al., 2021). For example, Safran et al. (2021) show that the condition is satisfied with high probability over the trajectory of perturbed gradient descent on over-parameterized two-layer ReLU networks when initialized in a neighborhood of a global minimum. We also note that there exists other variants of this definition such as quasi-strong convexity (Necoara et al., 2019) or $(1, \mu)$-(strong) quasar convexity (Hinder et al., 2020), which are similar but slightly stronger.

### 4.1 ESCAPING FROM LOCAL MINIMA WITH A LARGE LEARNING RATE

We first state a lemma which is the key to proving Theorem 1. In particular, this lemma defines a set of criteria for the region $M$ around a minimum $\mathbf{x}^\dagger$ as well as the region around $M$, called $P(M)$, that ensures GD escapes from $M$ moving toward a different minimum $\mathbf{x}_\star$. To build further intuition, Figure 3 provides an example of how GD with large learning rate may escape a sharp minimum. Figure 4 provides an illustration of different regions defined in the theorem's statement.

**Lemma 1.** *Let $f$ be a function that is $L_{global}$-smooth and consider running GD with learning rate $\gamma$ and randomly initialized over a set $W$ with $\mathcal{L}(W) > 0$. Let $M$ be a set with diameter $r$, containing a local minimum $\mathbf{x}^\dagger$ and define $P(M) := \{\mathbf{x} \notin M \mid \|\mathbf{x} - \mathbf{x}^\dagger\|_2 \leq r\sqrt{\gamma^2 L_{global}^2 - 3}\}$ to be the set surrounding $M$. Assume $f$ is $L < L_{global}$-smooth and $\mu_\star$-OPSC over $P(M)$ with respect to a (global) minimum $\mathbf{x}_\star$ that is sufficiently far from $M$, formally, $\|\mathbf{x}_\star - \mathbf{x}^\dagger\|_2 \geq r \cdot \frac{1 + \sqrt{(\gamma^2 L_{global}^2 - 3)(1 - \gamma\mu_\star)}}{1 - \sqrt{1 - \gamma\mu_\star}}$. Finally, assume $f$ is $\mu^\dagger$-OPSC with respect to $x^\dagger$ over $M$ where $\mu^\dagger > \frac{2L^2}{\mu_\star}$. Then, using a suitable learning rate $\frac{2}{\mu^\dagger} < \gamma \leq \frac{\mu_\star}{L^2}$, if GD reaches a point $M$, it will escape $M$ and reach a point with distance to $x_\star$ of less than $\|\mathbf{x}^\dagger - \mathbf{x}_\star\| - r$ almost surely.*

Note that this lemma allows for multiple local minima to exist on the landscape and only applies constraints around each local minimum. Furthermore, we point out that the lemma only ensures that GD will exit the local minima after some steps. In order to obtain convergence guarantees to the global minimum, it is necessary to assume a convergence property on the rest of the landscape as well. Indeed, this is how we build the class of functions to prove Theorem 1. We provide a complete proof of Lemma 1 in Appendix D and proceed by discussing some of our assumptions:

**OPSC condition inside $M$** We assume $f$ is OPSC with respect to a different minima inside $M$ in order to ensure GD will escape from $M$. However, other conditions might also ensure the same

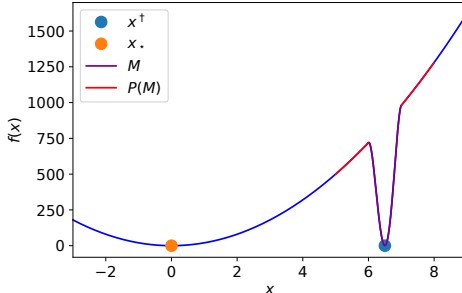
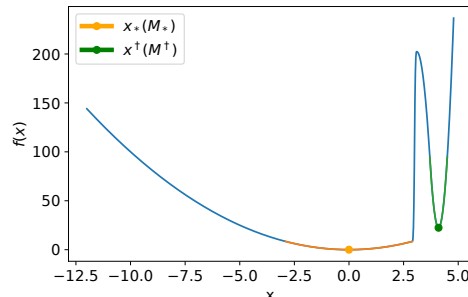

Figure 4: Illustration of different regions defined in Lemma 1.

Figure 5: An example where convergence to global minimum can only be achieved by using both stochastic noise and large learning rate but not with either used alone.

effect. The theorem would also hold with those assumptions. Note that the sharpness of $M$ with respect to the rest of landscape is reflected through the lower bound on $\mu^\dagger$ and is necessary so we can set the learning rate in the given range. As an example, when $f$ is a quadratic function everywhere except $M$ (such as in Figure 4), we have $\mu_\star = L$ and the lower bound becomes $\mu^\dagger > 2L$.

**OPSC condition around** $M$ We combine this assumption with the assumption on $M$ being sufficiently far from the global minimum in order to ensure that once GD escapes from a local minima, the gradient points strongly towards $\mathbf{x}_\star$. This ensures that GD will reach a point closer to the global minimum after escaping $M$. While the OPSC assumption is not necessary to show GD will never converge to $M$ and may be replaceable by alternatives, an assumption on the distance to $\mathbf{x}_\star$ might be necessary to show GD will not return to $M$. For example, consider a quadratic function where the region around minimum is replaced by a sharper quadratic function, as plotted in Figure 2. In this case, GD with a high learning rate will keep returning to $M$ though it will never converge to it. As alternatives to OPSC assumption on $P(M)$, one can assume GD converges in at most a fixed number of steps (which Corollary 1 states can not be to any point in $M$ almost surely) or assume directly that the gradient points strongly away from $M$. Finding similar assumptions is grounds for future work.

Given Lemma 1, it can be seen that if it is possible to ensure the iterations of GD get closer to the global minimum on the rest of the landscape, it is possible to ignore existence of the region $M$. This is because either the iterations would never cross $M$ or if they do, they will eventually reach a point closer to the global minimum according to Lemma 1. We build the class of functions for Theorem 1 based on this observation. We now state the formal statement of Theorem 1 and leave the proof to Appendix E.

**Theorem 1** (Formal). *Consider any function $f$ that is $L$-smooth and $\mu_\star$-OPSC with respect to the global minimum $\mathbf{x}_\star$ in its landscape except on a region $M$ containing a local minimum $\mathbf{x}^\dagger$ satisfying the conditions in Lemma 1. GD initialized randomly inside $M$ converges to $\mathbf{x}^\dagger$ with a small learning rate $\gamma < \frac{\mu^\dagger}{L_{global}^2}$. In contrast, GD initialized randomly over any arbitrary set $W$ with positive Lebesgue measure $\mathcal{L}(W) > 0$ will instead converge to $\mathbf{x}_\star$ with a large learning rate $\frac{2}{\mu^\dagger} < \gamma \le \frac{\mu_\star}{L^2}$ almost surely.*

## 4.2 Avoiding Local Minima

The following key lemma is used to prove Theorem 2.

**Lemma 2.** *Assume gradient descent is initialized randomly on the set $W$ and is run with learning rate $\gamma \le \frac{1}{2L}$. Let $X \in \mathcal{R}^d$ be an arbitrary set of points in the landscape. Assume $f$ is $L$-smooth over $\mathcal{R}^d \setminus X$. Let $\mathcal{L}(S)$ denote the Lebesgue measure of any set $S$. The probability of encountering any points of $X$ in the first $T$ steps of gradient descent, i.e. $\mathbf{x}_i \in X$ for some $1 \le i \le T$ is at most $2^{(T+1)\cdot d} \cdot \frac{\mathcal{L}(X)}{\mathcal{L}(W)}$.*

We provide the complete proof in Appendix B. Note that the lemma makes no assumption on $f$ over $X$. Furthermore, the dependence of the bound on $T$ seems inevitable in general since the optimization might force the iterations toward certain regions, such as the region around the minimum.

When using standard GD, by making assumptions on the landscape to ensure the iterations of GD move away from the undesired region $X$, the bound can be instead adapted to depend inversely on the learning rate which controls the speed of progression. We use one such assumption to prove Theorem 2 which we state formally here and leave its proof to Appendix C.

**Theorem 2** (Formal). *Let $X$ be an arbitrary set of points. Assume $f$ is L-smooth and $\mu_\star$-OPSC with respect to a minima $x_\star \notin X$ over $\mathbb{R}^d \setminus X$. Define $c_X := \inf\{\|\mathbf{x} - \mathbf{x}_\star\| \mid \mathbf{x} \in X\}$ and $r_W := \sup\{\|\mathbf{x} - \mathbf{x}_\star\| \mid \mathbf{x} \in W\}$. The probability of not encountering any points of $X$ during running gradient descent with $\gamma \leq \frac{\mu_\star}{L^2}$ is at least $1 - 2^d \cdot \frac{r_W}{c_X}^{-\frac{d}{\log_2(1 - \gamma\mu_\star)}} \cdot \frac{\mathcal{L}(X)}{\mathcal{L}(W)}$ when $c_X \leq r_W$ and is 1 otherwise.*

### 4.3 IMPORTANCE OF LARGE LEARNING RATE DESPITE THE EFFECTS OF STOCHASTIC NOISE

We now consider the case of SGD where the stochastic noise is applied as an additive term to the gradient. The update step of SGD in this case would be:

$$\mathbf{x}_{t+1} := \mathbf{x}_t - \gamma(\nabla f(\mathbf{x}_t) + \boldsymbol{\xi}_t). \tag{3}$$

For example, when the global objective $f(\mathbf{x})$ is a finite sum of $n$ different objectives, e.g. one for each data point, we will have $\boldsymbol{\xi}_t := \nabla f_{r_t}(\mathbf{x}) - \nabla f(\mathbf{x})$ where $r_t$ is the index of data point used in $t$-th step.

In this section we consider the case where the noise $\boldsymbol{\xi}_t$ is drawn from a uniform distribution $Uniform(-\sigma, \sigma)$ and assess the convergence point of GD on an example function plotted in Figure 5. This function contains a local minima $x^\dagger$ and a global minimum $x_\star$. Optimally, we would like to ensure convergence to the global minimum regardless of the initialization point. We now show that this is not possible using a small learning rate regardless of the magnitude of the noise. However, we show that when using a large enough learning rate alongside the stochastic noise, it is possible to obtain the desired result. We state this in the following proposition and leave a more formal description and its proof to Appendix G.

**Proposition 3.** *Consider running SGD on the function plotted in Figure 5. If the learning rate is sufficiently small, starting close to $x^\dagger$, the iterates will never converge to the optimum $x_\star$ nor to a small region around it regardless of the magnitude of the noise. On the other hand, by using a large learning rate, given that the stochastic noise satisfies certain bounds, GD will succeed to converge to the optimum $x_\star$ given any starting point.*

## 5 EXPERIMENTS

We now provide practical evidence to show the effects of high learning rate also apply and are essential in optimization of neural networks. In our experiments we train a ResNet-18 (He et al., 2016) without batch normalization on CIFAR10 (Krizhevsky & Hinton, 2009) dataset.

### 5.1 DISENTANGLING EFFECTS OF STOCHASTIC NOISE AND LEARNING RATE

As can be seen from (3), reducing the learning rate would also reduce the variance of the effective stochastic noise $\gamma\boldsymbol{\xi}_t$. This entanglement makes it hard to assess the effects of stochastic noise and large learning rate separately. We design the following method to maintain the level of noise when reducing the learning rate.

**SGD with Repeats**  In order to simulate the same magnitude of noise while still using a smaller learning rate, every time a mini-batch is drawn, we use it for $k$ steps before drawing another mini-batch. Note that when $k = 1$, we recover standard SGD. Re-using the same batch $k$ times, allows the bias of the mini-batch to be amplified $k$ times, so when reducing learning rate by $\frac{1}{k}$ the overall magnitude remains unchanged. This is explained in more detail in Appendix F.

We compare standard SGD with learning rate 0.01, standard SGD with learning rate 0.001, and SGD with $k = 10$ and learning rate 0.001. We apply 0.0005 weight decay, 0.9 momentum, and decay the learning rate at epochs 80, 120, and 160 by 0.1. Results without momentum are reported in Appendix M. When training with standard SGD and learning rate 0.001 we train the model for 10 times more epochs (2000 epochs) in order to obtain a fair comparison and rescale its plot to 200 epochs. In this case, learning rate decay happens at epochs 800, 1200, and 1600. Note that

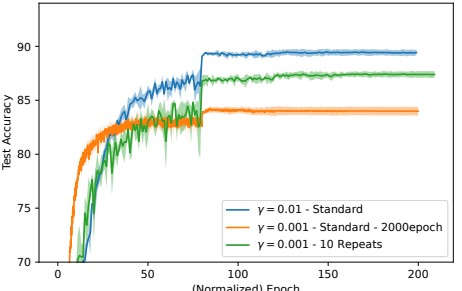 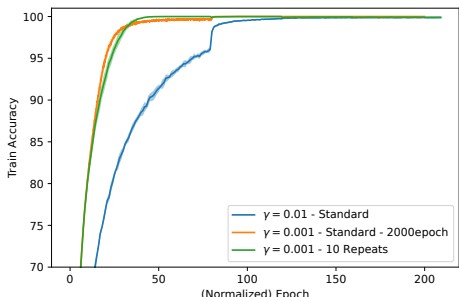

Figure 6: Comparsion between performance of SGD with different learning rates. The gap in performance between large and small learning rates, even after repeatedly using the same batch to maintain the effect of stochastic noise, suggests that learning rate has an effect on trajectory beyond controlling stochastic noise. Repeating batches is turned off at epoch 200 and 10 additional epochs are performed (green). For the experiment with 2000 epochs (orange), the plot is normalized to 200 epochs.

when running SGD with $k = 10$ repeats, we perform 10 steps using each batch while going through the whole dataset at each epoch. Therefore, the total number of steps in SGD with $k = 10$ is the same as standard SGD with 2000 epochs. Furthermore, when we have $k > 1$ we train the model for 10 more epochs at the end and use each batch only once (as in standard SGD) during the additional epochs. We perform these additional steps since training for several steps on one batch at the end of training might lead to overfitting on that batch which is not desirable for test performance. In Appendix K we also experiment with turning off repeats earlier in the training and observe no significant improvement. Finally, we ensure that the experiment with $k = 10$ uses the same initialization point and the same ordering of batches used for training with learning rate 0.01.

The results (averaged over 3 runs) are plotted in Figure 6. The first clear observation is that SGD with learning rate 0.01 leads to a much better model than SGD with learning rate 0.001. More importantly, while amplifying the noise through repeats helps lower the gap, it still has a performance below training with the large learning rate.

Explaining the positive effect of using SGD over GD on convergence has been the focus of several prior work. For example, Kleinberg et al. (2018) argue that applying SGD allows optimization to be done over a smoothed version of the function which empirically satisfies good convergence properties, particularly, one-point strong convexity toward a minimum. We argue that our observation provides a more complete overview and suggests that even after applying stochastic noise (which for example can lead to a smoothing of the function), there might be certain regions of the landscape that can only be avoided using a high learning rate. As we described above, one can consider the effect of stochastic noise to be the improvement observed when using repeats with a small learning rate in comparison with training in a standard way which still does not close the gap with training using a high learning rate. Therefore, the effects of using a high learning rate, such as those described in Section 4, are still important in determining the optimization trajectory even in stochastic setting.

## 5.2 COMPARING TRAJECTORIES OF LARGE AND SMALL LEARNING RATES

In Section 4, we proved some of the effects of using large step sizes in avoiding or escaping certain minima in the landscape. We now demonstrate that these effects can be observed in real world applications such as training neural networks. To be able to observe the effect of large learning rate more clearly, we first warm-start the optimization by running SGD with a small learning rate 0.001 with $k = 10$ repeats (as described in Section 5.1) for 20 epochs to obtain parameters $\mathbf{x}_{\text{warm}}$. We do this to get near a minimum that would be found when using the small learning rate. Then, we start full-batch GD from $\mathbf{x}_{\text{warm}}$ with two different learning rate 0.001 (small) and 0.01 (large). We do not apply momentum when performing full-batch GD but apply 0.0005 weight decay. However we did not observe any visible difference in the results without weight decay. Similar to (Li et al., 2018), we obtain the first two principal components of the vectors $\mathbf{x}_1 - \mathbf{x}_0, \mathbf{x}_2 - \mathbf{x}_0, \dots, \mathbf{x}_t - \mathbf{x}_0$ and plot the trajectory along these two directions in Figure 7. We can clearly observe that GD with a large learning rate changes path and moves toward a different place in the landscape. GD continues on the different path even when the learning rate is reduced back to 0.001 after 400 steps. Furthermore, looking at the loss values, we can observe a peak at the beginning of training that closely resembles

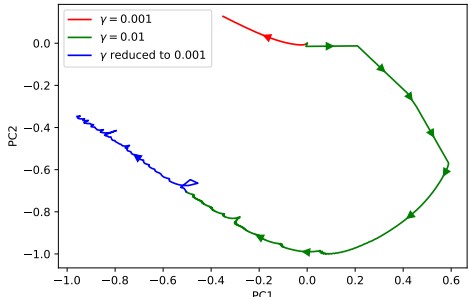 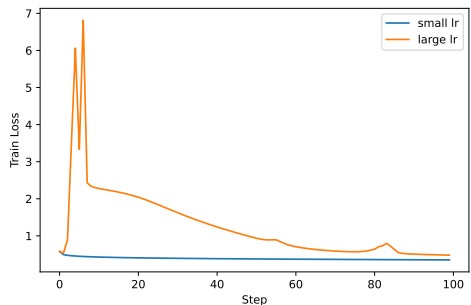

(a) Trajectory of (full-batch) GD with small and large learning rate.

(b) The value of train loss when using different values of learning rate.

Figure 7: Behavior of GD for learning rates 0.001 (small) and 0.01 (large). The initialization is obtained by warm-starting the network using SGD with a small learning rate 0.001. Using a large learning rate changes the trajectory sharply and even if the learning rate is reduced again after several steps (blue line) we move toward a different direction in the landscape. This is accompanied by a sharp increase of loss at the beginning that can be attributed to GD escaping from a local sharp region in the landscape.

what we expect to observe when GD is escaping from a local sharp region. This clearly shows that these behaviors of GD are not merely theoretical and are also relevant in real world applications.

Note that while we do not observe similar high spikes in the loss in the next steps, we conjecture that this behavior of escaping sharp regions is constantly happening throughout training. This is also confirmed by observations in (Cohen et al., 2021) which show sharpness increases throughout training until reaching the threshold $\frac{2}{\mu}$ where $\mu$ is the learning rate. GD will then oscillate between being sharper and smoother than this threshold. As a result of constantly avoiding sharp regions, symptoms of an escape such as a spike in loss is not observed. While we observe smaller increases in the loss, these can be due to oscillations also observed in (Cohen et al., 2021) along the highest eigenvectors which are not the same as escaping. Results in the same work show that in these cases the parameters do not move in these directions and only oscillate around the same center. Developing better visualization techniques or identifying other effects of using a high learning rate on GD's trajectory can help explain this behavior further and both of these directions are ground for future work.

## 6 FUTURE WORK

Obtaining further insight on how GD avoids locally sharp regions, for example by developing better methods for visualization of the landscape and trajectory, is grounds for future work. Furthermore, there are various extensions possible on the theoretical results obtained in this paper. For example, it might be possible to show other effects of using a large learning rate on trajectory that facilitate escaping from local minima. Finally, obtaining similar results with a relaxed set of assumptions would also be an interesting direction of research.

## 7 CONCLUSION

In this paper, we highlight that a high learning rate can provably lead to avoiding or escaping local minima and reaching a global minimum. Based on this result, we argue that analyzing GD with infinitesimally small learning rate is not sufficient to understand its success unlike what was suggested in prior work (Elkabetz & Cohen, 2021). Furthermore, by designing a method to amplify stochastic noise without increasing the learning rate, we disentangle the effects of stochastic noise and high learning rates. We observe that while a higher stochastic noise leads to a better model, it is not enough to close the gap with the model obtained using a high learning rate. Therefore, we argue that the effect of learning rate goes beyond controlling the impact of stochastic noise even in SGD. In contrast, recent works on analyzing success of SGD focus on continuous settings (Xie et al., 2021) and only take step size into account when modeling the noise (Pesme et al., 2021). We hope that our results will encourage future work on large step size regime. Finally, we demonstrate that the escape from sharp regions can happen in training of neural networks, hence signifying the relevance of the effects of large learning rate in real world applications.

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

# A    GRADIENT DESCENT WITH RANDOM INITIALIZATION

---

**Algorithm 1** Gradient Descent with Random Initialization

---
1: Pick $\mathbf{x}_0$ randomly from the set $W$.
2: **for** $t = 1 \ldots T$ **do**
3:      $\mathbf{x}_t \leftarrow \mathbf{x}_{t-1} - \gamma \nabla f(\mathbf{x}_{t-1})$
4: **end for**

---

# B    PROOF OF LEMMA 2

**Lemma.** *Assume gradient descent is initialized randomly on the set $W$ and is run with learning rate $\gamma \leq \frac{1}{2L}$. Let $X \in \mathcal{R}^d$ be the set of points that should not be encountered by GD and assume $f$ is $L$-smooth over $\mathcal{R}^d \setminus X$. Let $\mathcal{L}(S)$ denote the Lebesgue measure of any set $S$. The probability of encountering any points of $X$ in the first $T$ steps of gradient descent, i.e. $\mathbf{x}_i \in X$ for some $1 \leq i \leq T$ is at most $2^{(T+1)\cdot d} \cdot \frac{\mathcal{L}(X)}{\mathcal{L}(W)}$.*

*Proof.* Define $g(\mathbf{x}) := \mathbf{x} - \gamma \nabla f(\mathbf{x})$. When $\gamma < \frac{1}{L}$, since $f$ is $L$-smooth over $\mathcal{D}_g := \mathcal{R}^d \setminus X$, results of Lee et al. (2016) show $g(\mathbf{x})$ is a diffeomorphism over $\mathcal{D}_g$. As a result, the function $g^T$ obtained by applying $g$ for $T$ times is also a diffeomorphism over the set

$$\mathcal{D}_{g^T} := \mathcal{R}^d \setminus (X \cup g^{-1}(X) \cup \ldots \cup (g^{(T-1)})^{-1}(X)).$$

According to the change of variable formula for Lebesgue measure (for example, see (Bogachev, 2006, Eq. (3.7.2))), for any measurable set $Y \subset \mathcal{D}_{g^T}$

$$\mathcal{L}(g^T(Y)) = \int_Y |\det \nabla g^T(\mathbf{y})| \mathbf{dy} \,.$$

Since $\gamma \leq \frac{1}{2L}$, we have for any $\mathbf{y} \in \mathcal{D}_g$,

$$|\det \nabla g(\mathbf{y})| = |\det(I - \gamma \nabla^2 f(\mathbf{y}))| \geq 2^{-d} \,.$$

The last equality holds because smoothness ensures all eigenvalues of $\nabla^2 f(\mathbf{x})$ are at most $L$. So for any eigenvalue $\lambda_i$, $1 - \gamma \lambda_i \geq \frac{1}{2}$. Using this result, we also can obtain $|\det \nabla g^T(\mathbf{y})| \geq 2^{-Td}$ for any $\mathbf{y} \in \mathcal{D}_{g^T}$. Thus, we have

$$\mathcal{L}(g^T(Y)) \geq 2^{-Td} \mathcal{L}(Y) \,,$$

which means,

$$\mathcal{L}((g^T)^{-1}(X) \cap \mathcal{D}_{g^T}) \leq 2^{Td} \mathcal{L}(X) \,.$$

Note that while the above argument works for $T \geq 1$, the former inequality also trivially holds for $T = 0$. Hence

$$
\begin{aligned}
\mathcal{L}(\cup_{t=0}^T ((g^t)^{-1}(X) \cap W)) &\leq \mathcal{L}(\cup_{t=0}^T (g^t)^{-1}(X)) \\
&= \mathcal{L}(\cup_{t=0}^T ((g^t)^{-1}(X) \cap \mathcal{D}_{g^t})) \\
&\leq \sum_{t=0}^T \mathcal{L}((g^t)^{-1}(X) \cap \mathcal{D}_{g^t}) \\
&\leq \sum_{t=0}^T 2^{td} \mathcal{L}(X) \\
&\leq \mathcal{L}(X) (\sum_{t=0}^T 2^t)^d \\
&\leq 2^{(T+1)d} \mathcal{L}(X) \,,
\end{aligned}
$$

where in the last inequality we used $2^0 + 2^1 + \ldots 2^T < 2^{T+1}$. The theorem follows directly from this result. $\qquad\square$

The following corollary directly follows from Lemma 2. We use this corollary in proving Lemma 1 to avoid cases where we directly land on a minimum with $\nabla f(\mathbf{x}) = 0$.

**Corollary 1.** *Let $f$ be $L$ smooth. If $X$ is a set with $\mathcal{L}(X) = 0$, for example when it is a finite set of points, the probability of encountering $X$ throughout training with gradient descent using $\gamma \leq \frac{1}{2L}$ and random initialization over a set $W$ with $\mathcal{L}(W) > 0$ is 0.*

## C  PROOF OF THEOREM 2

**Theorem.** *Let $f$ be $L$-smooth and $\mu_\star$-OPSC with respect to a minima $x_\star$ over $\mathbb{R}^d \setminus X$. Define $c_X := \inf\{\|\mathbf{x} - \mathbf{x}_\star\| \mid \mathbf{x} \in X\}$ and $r_W := \sup\{\|\mathbf{x} - \mathbf{x}_\star\| \mid \mathbf{x} \in W\}$. The probability of encountering any points of $X$ during running gradient descent with $\gamma \leq \frac{\mu_\star}{L^2}$ is upper bounded by $2^d \cdot \frac{r_W}{c_X}^{-\frac{d}{\log_2(1-\gamma\mu_\star)}} \cdot \frac{\mathcal{L}(X)}{\mathcal{L}(W)}$ when $c_X \leq r_W$ and is zero otherwise.*

*Proof.* Due to $\mu_\star$-OPSC property of the landscape over $\mathbb{R}^d \setminus X$, as long as $\mathbf{x}_t \notin X$, we have

$$
\begin{aligned}
\|\mathbf{x}_{t+1} - \mathbf{x}_\star\|_2^2 &= \|\mathbf{x}_t - \gamma\nabla f(\mathbf{x}_t) - \mathbf{x}_\star\|_2^2 \\
&= \|\mathbf{x}_t - \mathbf{x}_\star\|_2^2 - 2\gamma \langle \nabla f(\mathbf{x}_t), \mathbf{x}_t - \mathbf{x}_\star \rangle + \gamma^2 \|\nabla f(\mathbf{x}_t)\|_2^2 \\
&\leq (1 - 2\gamma\mu_\star + \gamma^2 L^2)\|\mathbf{x}_t - \mathbf{x}_\star *\|_2^2 \\
&\leq (1 - \gamma(2\mu_\star - \gamma L^2))\|\mathbf{x}_t - \mathbf{x}_\star\|_2^2 \\
&\leq (1 - \gamma\mu_\star)\|\mathbf{x}_t - \mathbf{x}_\star\|_2^2,
\end{aligned}
$$

where the last inequality holds because $\gamma \leq \frac{\mu_\star}{L^2}$. Hence, if $\mathbf{x}_t \notin X$ for $t \in [T-1]$, we have

$$
\begin{aligned}
\|\mathbf{x}_T - \mathbf{x}_\star\|_2^2 &\leq (1 - \gamma\mu_\star)^T \|\mathbf{x}_t - \mathbf{x}_\star\|_2^2 \\
&\leq (1 - \gamma\mu_\star)^T r_W.
\end{aligned}
$$

Let $T_0 := \frac{\log_2 \frac{c_X}{r_W}}{\log_2 (1-\gamma\mu_\star)}$. For $T > T_0$, we have

$$
\|\mathbf{x}_T - \mathbf{x}_\star\|_2^2 \leq (1 - \gamma\mu_\star)c_X < c_X,
$$

which means $\mathbf{x}_T \notin X$. Therefore, if GD does not reach any point in $X$ in the first $T_0$ steps, it will not reach any point in $X$ afterwards neither. Therefore, the probability of encountering any point in $X$ is the same as the probability of encountering such points in the first $T_0$ steps. According to Lemma 2, this value is bounded as:

$$
\begin{aligned}
2^{(T_0+1)d} \cdot \frac{\mathcal{L}(X)}{\mathcal{L}(W)} &= 2^d \cdot \frac{c_X}{r_W}^{\frac{d}{\log_2 (1-\gamma\mu_\star)}} \cdot \frac{\mathcal{L}(X)}{\mathcal{L}(W)} \\
&= 2^d \cdot \frac{r_W}{c_X}^{-\frac{d}{\log_2 (1-\gamma\mu_\star)}} \cdot \frac{\mathcal{L}(X)}{\mathcal{L}(W)}. \qquad \square
\end{aligned}
$$

## D  PROOF OF LEMMA 1

**Lemma.** *Let $f$ be a function that is $L_{global}$-smooth and consider running GD with learning rate $\gamma$ randomly initialized over a set $W$ with $\mathcal{L}(W) > 0$. Let $M$ be a set with diameter $r$, containing a local minimum $\mathbf{x}^\dagger$ and define $P(M) := \{\mathbf{x} \notin M \mid \|\mathbf{x} - \mathbf{x}^\dagger\|_2 \leq r\sqrt{\gamma^2 L_{global}^2 - 3}\}$ to be the set surrounding $M$. Assume $f$ is $L < L_{global}$-smooth and $\mu_\star$-OPSC over $P(M)$ with respect to a (global) minimum $\mathbf{x}_\star$ that is sufficiently far from $M$, formally, $\|\mathbf{x}_\star - \mathbf{x}^\dagger\|_2 \geq r \cdot \frac{1+\sqrt{(\gamma^2 L_{global}^2 - 3)(1-\gamma\mu_\star)}}{1-\sqrt{1-\gamma\mu_\star}}$. Finally, assume $f$ is $\mu^\dagger$-OPSC with respect to $x^\dagger$ over $M$ where $\mu^\dagger > \frac{2L^2}{\mu_\star}$. Then, using a suitable learning rate $\frac{2}{\mu^\dagger} < \gamma \leq \frac{\mu_\star}{L^2}$, if GD reaches a point $M$, it will escape $M$ and reach a point with distance to $x_\star$ of less than $\|\mathbf{x}^\dagger - \mathbf{x}_\star\| - r$ almost surely.*

*Proof.* Let $t$ be the smallest step where $\mathbf{x}_t \in M$. Using Corollary 1, $\mathbf{x}_t \neq \mathbf{x}^\dagger$ almost surely. Therefore $\|\mathbf{x}_t - \mathbf{x}^\dagger\| > 0$. Since $\gamma > \frac{2}{\mu^\dagger}$, we have

$$
\begin{aligned}
\|\mathbf{x}_{t+1} - \mathbf{x}^\dagger\|_2^2 &= \|\mathbf{x}_t - \gamma \nabla f(\mathbf{x}_t) - \mathbf{x}^\dagger\|_2^2 \\
&= \|\mathbf{x}_t - \mathbf{x}^\dagger\|_2^2 - 2\gamma \langle \nabla f(\mathbf{x}_t), \mathbf{x}_t - \mathbf{x}^\dagger \rangle + \gamma^2 \|\nabla f(\mathbf{x}_t)\|_2^2 \\
&\geq \|\mathbf{x}_t - \mathbf{x}^\dagger\|_2^2 - 2\gamma \|\nabla f(\mathbf{x}_t)\|_2 \|\mathbf{x}_t - \mathbf{x}^\dagger\|_2 + \gamma^2 \|\nabla f(\mathbf{x}_t)\|_2^2 \\
&= \|\mathbf{x}_t - \mathbf{x}^\dagger\|_2^2 + \gamma \|\nabla f(\mathbf{x}_t)\|_2 (\gamma \|\nabla f(\mathbf{x}_t)\|_2 - 2\|\mathbf{x}_t - \mathbf{x}^\dagger\|_2) \\
&\geq \|\mathbf{x}_t - \mathbf{x}^\dagger\|_2^2 + \gamma \|\nabla f(\mathbf{x}_t)\|_2 (\gamma \mu^\dagger \|\mathbf{x}_t - \mathbf{x}^\dagger\|_2 - 2\|\mathbf{x}_t - \mathbf{x}^\dagger\|_2) \\
&\geq \|\mathbf{x}_t - \mathbf{x}^\dagger\|_2^2 + \gamma \|\nabla f(\mathbf{x}_t)\|_2 \|\mathbf{x}_t - \mathbf{x}^\dagger\|_2 (\gamma \mu^\dagger - 2) \\
&\overset{(A)}{\geq} \|\mathbf{x}_t - \mathbf{x}^\dagger\|_2^2 + \gamma \mu^\dagger \|\mathbf{x}_t - \mathbf{x}^\dagger\|_2^2 (\gamma \mu^\dagger - 2) \\
&\overset{(B)}{\geq} (2\gamma \mu^\dagger - 3) \|\mathbf{x}_t - \mathbf{x}^\dagger\|_2^2 ,
\end{aligned}
$$

where (A) holds because $\gamma \mu^\dagger - 2 > 0$ and (B) is obtained by using the lower bound $\gamma \mu^\dagger > 2$. Therefore, the distance to $\mathbf{x}^\dagger$ grows at least with the rate $2\gamma \mu^\dagger - 3 > 1$. Hence, GD is guaranteed to reach a point $\mathbf{x}_{t+k}$ outside $M$ for some $k > 0$. If $\|\mathbf{x}_{t+k} - \mathbf{x}_\star\| \leq \|\mathbf{x}^\dagger - \mathbf{x}_\star\| - r$, we are done. Otherwise, we verify that this condition holds for $\mathbf{x}_{t+k+1}$.

First note that

$$
\begin{aligned}
\|\mathbf{x}_{t+k} - \mathbf{x}^\dagger\|_2^2 &= \|\mathbf{x}_{t+k-1} - \gamma \nabla f(\mathbf{x}_{t+k-1}) - \mathbf{x}^\dagger\|_2^2 \\
&= \|\mathbf{x}_{t+k-1} - \mathbf{x}^\dagger\|_2^2 - 2\gamma \langle \nabla f(\mathbf{x}_{t+k-1}), \mathbf{x}_{t+k-1} - \mathbf{x}^\dagger \rangle + \gamma^2 \|\nabla f(\mathbf{x}_{t+k-1})\|_2^2 \\
&\leq \|\mathbf{x}_{t+k-1} - \mathbf{x}^\dagger\|_2^2 (1 - 2\gamma \mu^\dagger + \gamma^2 L_{\text{global}}^2) \\
&\leq r^2 (1 - 2\gamma \mu^\dagger + \gamma^2 L_{\text{global}}^2) \\
&\leq r^2 (\gamma^2 L_{\text{global}}^2 - 3) ,
\end{aligned}
$$

where the last inequality holds because $\gamma > \frac{2}{\mu^\dagger}$.

$$
\begin{aligned}
\|\mathbf{x}_{t+k+1} - \mathbf{x}_\star\|_2^2 &= \|\mathbf{x}_{t+k} - \gamma \nabla f(\mathbf{x}_{t+k}) - \mathbf{x}_\star\|_2^2 \\
&= \|\mathbf{x}_{t+k} - \mathbf{x}_\star\|_2^2 - 2\gamma \langle \nabla f(\mathbf{x}_{t+k}), \mathbf{x}_{t+k} - \mathbf{x}_\star \rangle + \gamma^2 \|\nabla f(\mathbf{x}_{t+k})\|_2^2 \\
&\leq \|\mathbf{x}_{t+k} - \mathbf{x}_\star\|_2^2 (1 - 2\gamma \mu_\star + \gamma^2 L^2) \\
&\leq \|\mathbf{x}_{t+k} - \mathbf{x}_\star\|_2^2 (1 - \gamma \mu_\star) ,
\end{aligned}
$$

where in the last inequality we used $\gamma \leq \frac{\mu_\star}{L^2}$. We can now write

$$
\begin{aligned}
\|\mathbf{x}_{t+k+1} - \mathbf{x}_\star\|_2 &\leq (\|\mathbf{x}^\dagger - \mathbf{x}_\star\|_2 + \|\mathbf{x}_{t+k} - \mathbf{x}^\dagger\|_2) \sqrt{1 - \gamma \mu_\star} \\
&\leq \left( \|\mathbf{x}^\dagger - \mathbf{x}_\star\|_2 + r\sqrt{\gamma^2 L_{\text{global}}^2 - 3} \right) \sqrt{1 - \gamma \mu_\star} .
\end{aligned}
$$

Given the lower bound on distance $\|\mathbf{x}^\dagger - \mathbf{x}_\star\|$, we have

$$
r(\sqrt{(\gamma^2 L_{\text{global}}^2 - 3)(1 - \gamma \mu_\star)} + 1) \leq \|\mathbf{x}^\dagger - \mathbf{x}_\star\|_2 (1 - \sqrt{1 - \gamma \mu_\star}) .
$$

This yields

$$
\|\mathbf{x}_{t+k+1} - \mathbf{x}_\star\|_2 \leq \|\mathbf{x}^\dagger - \mathbf{x}_\star\|_2 - r ,
$$

completing the proof. □

## E  PROOF OF THEOREM 1

Consider any function $f$ that is $L$-smooth and $\mu_\star$-OPSC with respect to some minimum $\mathbf{x}_\star$ in its landscape except on a region $M$ containing a local minimum $\mathbf{x}^\dagger$ satisfying the conditions in Lemma 1. GD initialized randomly inside $M$ converges to $\mathbf{x}^\dagger$ with a small learning rate $\gamma < \frac{\mu^\dagger}{L_{\text{global}}^2}$ but will instead converge to $\mathbf{x}_\star$ with a large learning rate $\frac{2}{\mu^\dagger} < \gamma \leq \frac{\mu_\star}{L^2}$ almost surely.

*Proof.* When GD is initialized inside $M$ and the learning rate is small satisfying $\gamma < \frac{\mu^\dagger}{L^2_{\text{global}}}$, since we have $\mu^\dagger$-OPSC and $L_{\text{global}}$-smoothness inside $M$, the iterates will satisfy

$$
\begin{aligned}
\|\mathbf{x}_{t+1} - \mathbf{x}^\dagger\|_2^2 &= \|\mathbf{x}_t - \gamma\nabla f(\mathbf{x}_t) - \mathbf{x}^\dagger\|_2^2 \\
&= \|\mathbf{x}_t - \mathbf{x}^\dagger\|_2^2 - 2\gamma\left\langle\nabla f(\mathbf{x}_t), \mathbf{x}_t - \mathbf{x}^\dagger\right\rangle + \gamma^2\|\nabla f(\mathbf{x}_t)\|_2^2 \\
&\leq (1 - 2\gamma\mu^\dagger + \gamma^2 L^2_{\text{global}})\|\mathbf{x}_t - \mathbf{x}^\dagger\|_2^2 \\
&\leq (1 - \gamma(2\mu^\dagger - \gamma L^2_{\text{global}}))\|\mathbf{x}_t - \mathbf{x}^\dagger\|_2^2 \\
&\leq (1 - \gamma\mu^\dagger)\|\mathbf{x}_t - \mathbf{x}^\dagger\|_2^2,
\end{aligned}
$$

Therefore, GD will converge to $\mathbf{x}^\dagger$. Let us now consider the case when GD is instead applied using a large learning rate satisfying $\frac{2}{\mu^\dagger} < \gamma \leq \frac{\mu_\star}{L^2}$. Furthermore, we allow initialization over any arbitrary set (instead of only subsets of $M$) as long as they satisfy $\mathcal{L}(W) > 0$. In this case, for each iterate, if $\mathbf{x}_t \notin M$, similar to above we have

$$
\|\mathbf{x}_{t+1} - \mathbf{x}_\star\|_2^2 \leq (1 - \gamma\mu_\star)\|\mathbf{x}_t - \mathbf{x}_star\|_2^2.
$$

If $\mathbf{x}_t \in M$, Lemma 1 shows that there exists $k > 0$ such that $\mathbf{x}_{t+k} \notin M$ and $\|\mathbf{x}_{t+k} - \mathbf{x}_\star\|_2^2$ is less than the distance of any point in $M$ to $\mathbf{x}_\star$. Since $\mathbf{x}_{t+k} \notin M$ the above argument holds and the distance to $\mathbf{x}_\star$ decreases. Therefore, for any $t' > t + k$ this distance $\|\mathbf{x}_{t'} - \mathbf{x}_\star\|_2^2$ remains less than the distance of any point in $M$ to $\mathbf{x}_\star$. This guarantees that $\mathbf{x}_{t'} \notin M$. Hence the distance to $\mathbf{x}_\star$ keeps decreasing which means GD will converge to $\mathbf{x}_\star$. $\square$

## F    EFFECT OF LEARNING RATE ON STOCHASTICITY

Let us focus on the case where $f(\mathbf{x})$ is the finite-sum $\frac{1}{N}\sum_{i=1}^N f_i(\mathbf{x})$. Then, using a large learning rate $k\gamma$, the iterates would satisfy

$$
\frac{\mathbf{x}_{t+1} - \mathbf{x}_t}{\gamma} = -k\nabla f_{r_t}(\mathbf{x}_t) = -k\nabla f(\mathbf{x}_t) - k(\nabla f_{r_t}(\mathbf{x}_t) - \nabla f(\mathbf{x}_t)). \tag{4}
$$

Let us assume that the deviation direction of each data point from the true gradient changes very slowly, i.e. the functions $f_i - f$ are extremely smooth. Then, using a smaller learning rate we instead have

$$
\begin{aligned}
\frac{\mathbf{x}_{t+k} - \mathbf{x}_t}{\gamma} &= -\sum_{i=0}^{k-1}\nabla f_{r_{t+i}}(\mathbf{x}_{t+i}) \\
&= -\sum_{i=0}^{k-1}\nabla f(\mathbf{x}_{t+i}) - \sum_{i=0}^{k-1}(\nabla f_{r_{t+i}}(\mathbf{x}_{t+i}) - \nabla f(\mathbf{x}_{t+i})) \\
&\approx -\sum_{i=0}^{k-1}\nabla f(\mathbf{x}_{t+i}) - \sum_{i=0}^{k-1}(\nabla f_{r_{t+i}}(\mathbf{x}_t) - \nabla f(\mathbf{x}_t)).
\end{aligned}
$$

To compare the strength of noise in each case we can for example compare the variance of the right hand side. Let $\sigma^2 := \frac{1}{N}\sum_{i=1}^N\|\nabla f_i(\mathbf{x}_t) - \nabla f(\mathbf{x}_t)\|_2^2$. Then, the variance when using the large learning rate would be $k^2\sigma^2$. When using a smaller learning rate and sampling at each step to obtain $r_t$ the variance is instead $k\sigma^2$ and is therefore reduced. However, using SGD with repeats, i.e. using $r_{t+i} = r_t$ for $1 \leq i \leq k-1$, we recover the same variance as the large learning rate. Therefore, using SGD with repeats, allows maintaining the same level of noise while still using a smaller learning rate.

## G    PROOF OF PROPOSITION 3

We first state the following key theorem which describes criteria ensuring escaping from or staying around a minimum:

**Theorem 4.** *Let $M$ be a ball with radius $r$ centered at a minimum $\mathbf{x}^\dagger$ and assume $f$ is $L_M$-smooth over $M$ and $\mu^\dagger$-OPSC with respect to $\mathbf{x}^\dagger$. Consider running SGD with a small learning rate $\gamma \leq \frac{\mu^\dagger}{2L_M^2}$. Assume that when running SGD such that the oracle noise is bounded as*

$$\mathbb{E}\|\mathbf{g}_t - \nabla f(\mathbf{x}_t)\|_2^2 \leq \sigma^2 \,.$$

*Furthermore assume that for some $c \leq 1$ we have $\Pr\left[\|\mathbf{g}_t - \nabla f(\mathbf{x}_t)\|_2^2 > c^2\sigma^2\right] > 0$ for all $\mathbf{x}_t \in M$. Assume SGD reaches a point in $M$. If $\sigma^2 \leq \frac{\mu^\dagger}{\gamma} r^2$ it will remain in $M$ with probability at least $\frac{2}{2-\gamma\mu^\dagger}$. On the other hand, if $c^2\sigma^2 \geq \frac{2L_M}{\gamma} r^2 + \epsilon$ for some $\epsilon > 0$ it will escape $M$ almost surely.*

*Proof.* Let $t$ denote the parameters at an iteration such that $\mathbf{x}_t \in M$. We have

$$
\begin{aligned}
\mathbb{E}\|\mathbf{x}_{t+1} - \mathbf{x}^\dagger\|^2 &= \|\mathbf{x}_t - \mathbf{x}^\dagger\|^2 - 2\gamma \left\langle \mathbb{E}\mathbf{g}_t, \mathbf{x}_t - \mathbf{x}^\dagger \right\rangle + \gamma^2 \mathbb{E}\|\mathbf{g}_t\|^2 \\
&= \|\mathbf{x}_t - \mathbf{x}^\dagger\|^2 - 2\gamma \left\langle \nabla f(\mathbf{x}_t), \mathbf{x}_t - \mathbf{x}^\dagger \right\rangle + \gamma^2 \|\nabla f(\mathbf{x}_t)\|^2 + \gamma^2\sigma^2 \\
&\leq \|\mathbf{x}_t - \mathbf{x}^\dagger\|^2 (1 - 2\gamma\mu^\dagger + \gamma^2 L_M^2) + \gamma r^2 \mu^\dagger \\
&\leq r^2 (1 - \gamma\mu^\dagger + \gamma^2 L_M^2) \\
&\leq r^2 (1 - \frac{\gamma\mu^\dagger}{2}) \,.
\end{aligned}
$$

Thus, using Markov inequality we have

$$\Pr\left[\|\mathbf{x}_{t+1} - \mathbf{x}^\dagger\|^2 > r^2\right] \leq \frac{1}{1 - \frac{\gamma\mu^\dagger}{2}} \,,$$

which shows the claim. On the other hand, let $p := \Pr\left[\|\mathbf{g}_t - \nabla f(\mathbf{x}_t)\|_2^2 > c^2\sigma^2\right]$. Then with probability at least $p > 0$ we have

$$
\begin{aligned}
\|\mathbf{x}_{t+1} - \mathbf{x}^\dagger\|_2^2 &= \|\mathbf{x}_t - \gamma\nabla f(\mathbf{x}_t) - \mathbf{x}^\dagger - \gamma(\mathbf{g}_t - \nabla f(\mathbf{x}_t))\|_2^2 \\
&= \|\mathbf{x}_t - \mathbf{x}^\dagger\|_2^2 - 2\gamma \left\langle \nabla f(\mathbf{x}_t), \mathbf{x}_t - \mathbf{x}^\dagger \right\rangle + \gamma^2 \|\nabla f(\mathbf{x}_t)\|_2^2 + \gamma^2 \|\mathbf{g}_t - \nabla f(\mathbf{x}_t)\|_2^2 \\
&\geq \|\mathbf{x}_t - \mathbf{x}^\dagger\|_2^2 - 2\gamma \|\nabla f(\mathbf{x}_t)\|_2 \|\mathbf{x}_t - \mathbf{x}^\dagger\|_2 + \gamma^2 \|\nabla f(\mathbf{x}_t)\|_2^2 + c^2\gamma^2\sigma^2 \\
&= \|\mathbf{x}_t - \mathbf{x}^\dagger\|_2^2 + \gamma\|\nabla f(\mathbf{x}_t)\|_2 (\gamma\|\nabla f(\mathbf{x}_t)\|_2 - 2\|\mathbf{x}_t - \mathbf{x}^\dagger\|_2) + c^2\gamma^2\sigma^2 \\
&\geq \|\mathbf{x}_t - \mathbf{x}^\dagger\|_2^2 + \gamma\|\nabla f(\mathbf{x}_t)\|_2 (\gamma\mu^\dagger\|\mathbf{x}_t - \mathbf{x}^\dagger\|_2 - 2\|\mathbf{x}_t - \mathbf{x}^\dagger\|_2) + c^2\gamma^2\sigma^2 \\
&\geq \|\mathbf{x}_t - \mathbf{x}^\dagger\|_2^2 + \gamma\|\nabla f(\mathbf{x}_t)\|_2 \|\mathbf{x}_t - \mathbf{x}^\dagger\|_2 (\gamma\mu^\dagger - 2) + c^2\gamma^2\sigma^2 \\
&\geq \|\mathbf{x}_t - \mathbf{x}^\dagger\|_2^2 + \gamma L_M \|\mathbf{x}_t - \mathbf{x}^\dagger\|_2^2 (\gamma\mu^\dagger - 2) + c^2\gamma^2\sigma^2 \\
&\geq \|\mathbf{x}_t - \mathbf{x}^\dagger\|_2^2 + \gamma L_M r^2 (\gamma\mu^\dagger - 2) + 2\gamma L_M r^2 + \epsilon \\
&\geq \|\mathbf{x}_t - \mathbf{x}^\dagger\|_2^2 + \gamma^2 L_M \mu^\dagger r^2 + \epsilon \,.
\end{aligned}
$$

This means the distance to $\mathbf{x}^\dagger$ grows at least with the constant $\epsilon$. Let $q := \frac{r^2}{\epsilon}$. Therefore, with probability at least $p^q$, one of $\mathbf{x}_{t+1}, \ldots, \mathbf{x}_{t+q}$ will be out of $M$. This holds for any consecutive $q$ iterates. Partitioning the iterates to parts of consecutive iterates of size $q$, each part has a positive probability of containing a point outside $M$. Therefore SGD will reach a point outside of $M$ almost surely. $\quad\square$

The function plotted in Figure 5 can be formally defined as follows:

$$
\begin{aligned}
f_1(x) &:= 86400((x - \alpha)^3 - (2.9 - \alpha)^3) + 2.9^2 \\
f_2(x) &:= \beta((x - 3.1)^3 + 0.001) + f_1(3) \\
f_3(x) &:= -300(x - 3.1)^2 + f_2(3.1) \\
f_{\text{tm}}(x) &:= \begin{cases}
x^2 & x \leq 2.9 \\
f_1(x) & 2.9 < x \leq 3 \\
f_2(x) & 3 < x \leq 3.1 \\
f_3(x) & 3.1 < x \leq 3.7 \\
450 * ((x - 4.1)^2 - 0.16) + f_3(3.7) & 3.7 < x
\end{cases}
\end{aligned}
$$

with $\alpha = 2.9 - \frac{\sqrt{2.9}}{360}$ and $\beta = 8640000(3 - \alpha)^2$. This function satisfies the following properties:

- $f_{\text{tm}}$ is 2-smooth over $\{x \mid x < 2.9\}$.
- $f_{\text{tm}}$ is 900-OPSC towards $4.1$ and 900-smooth over $\{x \mid 3.7 < x\}$.
- $f_{\text{tm}}$ is 4.5-OPSC towards $4.1$ over $\{x \mid 3.108 < x\}$.

We now proceed to proving Proposition 3, stating it formally here:

**Proposition 3.** *Consider running SGD on $f_{tm}$ with stochastic noise $\xi_t$ drawn i.i.d. at each step from the uniform distribution $Uniform(-\sigma, \sigma)$. If the learning rate is small such that it satisfies $\gamma < \frac{1}{30^2}$ the algorithm will not converge to $x_\star$ for some set of initialization points with positive Lebesgue measure. In contrast, with a large learning rate satisfying $0.4 \le \gamma \le 0.5$ it is possible to choose $\sigma$ such that the algorithm will converge to $x_\star$ almost surely.*

*Proof.* Assume $\gamma < \frac{1}{900}$. Consider the case where the algorithm is initialized inside $M_1 := \{x \mid 3.7 < x < 4.5\}$. Then if $\sigma$ satisfies $\sigma^2 \le \frac{900}{\gamma} \cdot 0.4^2$ it will remain in $M_1$ with positive probability according Theorem 4. According to the same theorem, If this bound is not satisfied, then we have $\frac{\sigma^2}{2} \ge \frac{2 \cdot 2 \cdot 2.9^2}{\gamma}$ which means any time SGD reaches a point in $M_\star := \{x \mid -2.9 < x < 2.9\}$, it will escape from it almost surely within a constant number of steps. This means that the algorithm will never stay close to $x_\star = 0$ forever or for an arbitrarily long number of steps.

Now consider the case where the learning rate is large enough. Choose $\sigma$ such that $5.1 < \sigma < 5.5$. Note that if the iterates reach the set $\{x \mid -2.9 < x < 2.9\}$ they will never exit it since we have

$$
\begin{aligned}
|x - \gamma(2x + \xi_t)| = |(1 - 2\gamma)x - \gamma\xi_t| \\
\le (1 - 2\gamma)(2.9) + \gamma(5.5) \\
\le (1 - 2\gamma)(2.9) + 2\gamma(2.9) \\
\le 2.9 \, .
\end{aligned}
$$

We will now show that from any other point there is a positive probability of reaching the range $[-2.9, 2.9]$. This fact combined with the almost sure guarantee of not escaping from $[-2.9, 2.9]$, proves that the algorithm will converge to this set almost surely.

Note that $f_{\text{tm}}$ is 4.5-OPSC towards $x = 4$ over the set $\{x \mid 3.108 < x\}$. Since $\gamma > 0.45$, it can be seen from the proof of Lemma 1 that SGD will continue to get further from $x = 4$ while it is in this set. Furthermore, given the direction of the gradients it is clear that the iterates would alternate between being less and more than $4$. Therefore, at some point, the iterates will exit this set reaching a point $x_t < 3.108$. Note that with a positive probability, the noise will not interfere with this escape since there is at least $0.5$ probability that the noise is aligned with the gradient direction.

If $3.1 < x < 3.108$, the gradient value is less than $5$. Since $\sigma > 5.1$ there is a positive probability of moving to the region $x < 3.1$. When $2.9 < x < 3.1$, because of the positive probability of alignment between the gradient and the noise, SGD will move to $x < 2.9$ with positive probability. Finally, given the smoothness of the region $x < 2.9$, if $x < -2.9$ SGD will converge toward $x_\star = 0$, ultimately reaching the region $-2.9 < x < 2.9$ with positive probability. This completes the proof. $\qquad\square$

## H  TOY EXAMPLE

In order to demonstrate the effects discussed in Section 4, we experiment with running GD over a simple function. The landscape of this function is plotted in Figure 8a and its formula is presented in Appendix I. The function has two minima, one near the initialization and one further away. Since the near-init minimum is almost completely flat, i.e. gradient is constant and equal to zero (except for the edges which are extremely sharp lines in order to ensure the function remains continuous), if GD reaches a point in this region, it will remain there. However, as this region is very close to the initialization, Lemma 2 (more particularly Corollary 2) suggests that GD with large enough learning rate, will probably not reach any points in this region. To demonstrate this more clearly, we plot the trajectory of GD from several initialization points in Figure 1. It is worth noting that even with large learning rate it is possible for GD to get stuck in this region while it is possible to avoid this

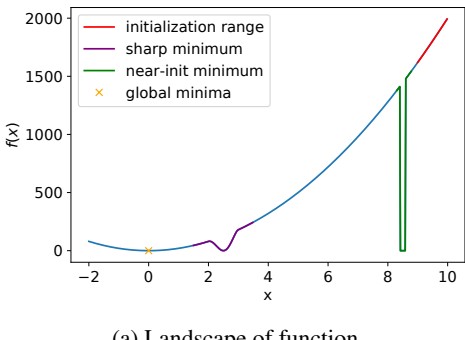
(a) Landscape of function

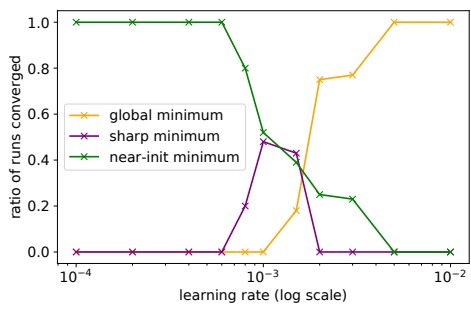
(b) Distribution of convergence to each minima

Figure 8: The function used in the toy experiments which has two local minima, a mostly flat minima near the initialization points and a sharp minima further away. It can be clearly observed how as the learning rate grows the two effects of avoiding parts of the landscape and escaping sharp minimas allow GD to converge to the global minimum.

region even with a small learning rate. However, as suggested by our theoretical upper bound, the probability of this phenomenon increases with the learning rate.

The other minimum is much sharper than the rest of the function and therefore we can expect an escaping behavior similar to the one described by Lemma 1. This behavior is demonstrated in Figure 3. Note that unlike the previous case, GD with large learning rate always (except when landing directly at the minimum) escapes the sharp minimum while GD with small learning rate converges.

We measure rate of convergence of GD for 100 different random initialization to each of these three regions for different learning rates. The results are plotted in Figure 8b. We observe that as the learning increases, the rate of avoiding the near initialization minimum increases. While the learning rate is not still high enough, GD will converge to the sharp minimum while as the learning rate increases further, it is also able to escape the sharp minimum and converge to the global minimum. This behavior is completely compatible with what can be expected based on the results and effects discussed in Section 4.

## I FUNCTION FOR TOY EXAMPLE

$$f(x) := \begin{cases} -1600(x-2.5)^5 - 2000(x-2.5)^4 + 800(x-2.5)^3 + 1020(x-2.5)^2 & 2 \le x \le 3, \\ 1411.2 \times (1 - 10^4(x-8.4)) & 8.4 \le x \le 8.40001, \\ 0 & 8.40001 \le x \le 8.59999, \\ 1479.2 \times (10^4(x-8.6) + 1) & 8.59999 \le x \le 8.6, \\ 20x^2 & \text{otherwise.} \end{cases}$$

## J 2D TOY EXAMPLE

To build more intuition and show the effect of large learning rate extends to multi-dimensions, we also provide a toy example on 2D. Figure 9 shows the landscape of our toy example which contains four local minima that are also sharp. Consider GD initialized randomly on the region $W := \{(x, y) \mid 3 \le x, y \le 4\}$. Then, using a small learning rate GD will converge to the minimum in the region $[1, 2] \times [1, 2]$. However, using a larger learning rate allows escaping that minimum. Increasing the magnitude, GD can also jump over the minimum completely. In these cases, GD will converge towards the global minimum at $(0, 0)$.

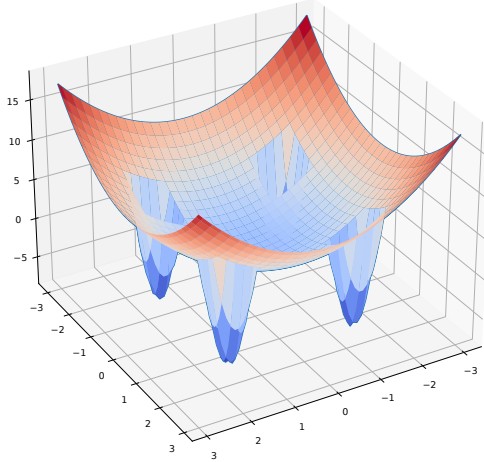

Figure 9: The landscape of the function $f(x, y) := x^2 + y^2 - 200ReLU(|x| - 1)ReLU(|y| - 1)ReLU(2 - |x|)ReLU(2 - |y|)$.

## K    RESULTS OF STOPPING REPEATS FROM DIFFERENT EPOCHS

In Section 5.1, we explained that at the end of training we stop using the same batch for $k$ steps and train in the standard way (each batch used just once) for additional 10 epochs. This was done to make sure the model that is used to obtain the accuracy on the test data is not overfitted on one batch which might be more likely to happen at the end of the training. In this section, we also experiment with stopping repeats, i.e. using the same batch for $k$ steps, earlier in the training. The result is plotted in Figure 10. No significant improvement is observed.

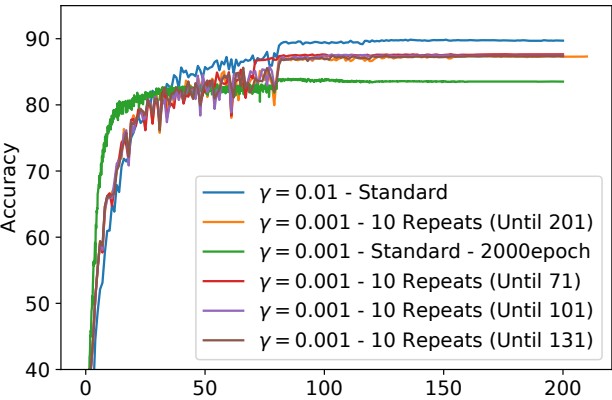

Figure 10: Plot of test accuracy when we stop using the same batch several times (doing repeats) at different epochs. It can be clearly observed that the stopping epoch does not affect the final accuracy and the gap with the case of GD with a large learning rate can be clearly observed.

## L    EXPERIMENTS ON CIFAR100

In order to make sure our results extend to other scenarios, we repeat the experiments in Section 5.1 on CIFAR100 and observe a similar behavior. The accuracy on the train and test datasets during training are plotted in Figure 11.

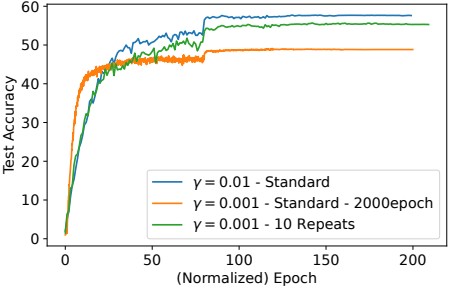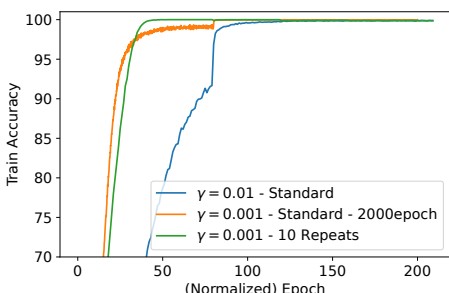

Figure 11: Comparsion between performance of SGD with different learning rates on CIFAR100. Repeating batches is turned off at epoch 200 and 10 additional epochs are performed (green). For the experiment with 2000 epochs (orange), the plot is normalized to 200 epochs. For more explanations refer to Figure 6 and Section 5.1.

## M    EXPERIMENTS ON SGD WITHOUT MOMENTUM

In Section 5.1, we designed an experiment to show the effect of large learning rate is important and goes beyond controlling the effect of stochastic noise on the trajectory. Since our goal was to demonstrate the relevance and importance of analyzing these effects for the practical scenarios, we used the standard training settings including momentum and weight decay. For completeness, in this section we also include the results of applying SGD with repeats without momentum and without weight decay. We compare standard SGD with learning rate 0.05. standard SGD with learning rate 0.005, and SGD with $k = 10$ repeats and learning rate 0.005. Accuracy on test and train datasets throughout training is plotted in Figure 12. The figure also contains the accuracy during training with momentum to allow comparison. As expected, applying SGD without momentum performs worse than SGD with momentum. The gap between small and large learning rate can be observed in this case as well. However, we do not observe an improvement when applying repeats.

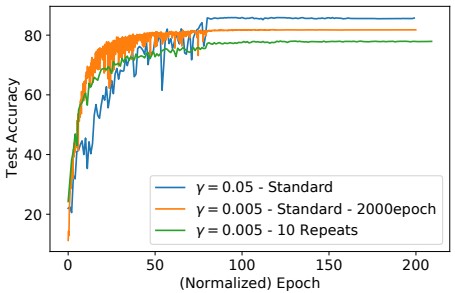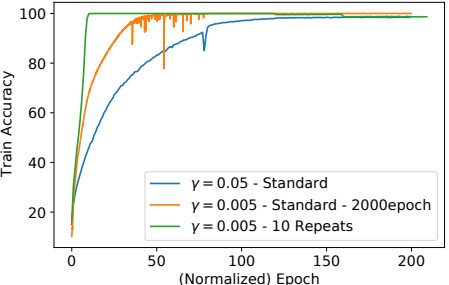

Figure 12: Comparsion between performance of SGD without momentum and weight decay and with different learning rates on CIFAR10. Repeating batches is turned off at epoch 200 and 10 additional epochs are performed (green). For the experiment with 2000 epochs (orange), the plot is normalized to 200 epochs. For more explanations refer to Figure 6 and Section 5.1.

## N    LOSS ON THE LINE BETWEEN LARGE AND SMALL LEARNING RATE TRAJECTORIES

In Section 5.2, we observed that GD with a large learning rate shows behavior similar to escaping and follows a different trajectory than GD with the small learning rate. In this section, we plot the loss along the line between the first point in the trajectory of GD with small learning rate (hereafter called the origin) and different points along the trajectory of GD with the large learning rate. Figure 13 shows the loss based on the norm of the distance to the origin. As expected the loss increases along the line between the origin and points at the beginning of the trajectory. This is when GD is showing

escaping behaviors. However, interestingly, the loss is decreasing along the line between the origin and points encountered later in the trajectory.

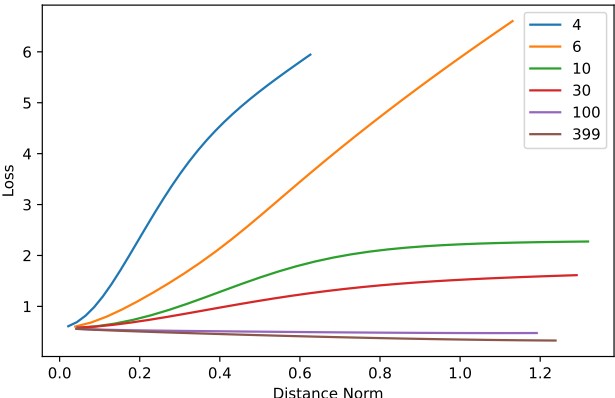

Figure 13: The value of loss along the line between the first point in the trajectory of GD with small learning rate and different points in the trajectory of GD with a large learning rate. For more detailed explanation of the settings, refer to Section 5.2. Each line corresponds to the value of loss measured on 30 points along the line between the initialization and the parameters after an step. The step number for each line is written in the box located on the top-right of the plot.

## ADDITIONAL REFERENCES

V.I. Bogachev. *Measure Theory*. Springer Berlin Heidelberg, 2006. ISBN 978-3-540-34513-8.

