# OpenReview forum: "Large Learning Rate Matters for Non-Convex Optimization"
_ICLR.cc/2023/Conference — Submitted to ICLR 2023_

### Official Review · Reviewer_hE2g · 2022-10-25

**Confidence:** 4
**Correctness:** 3
**Technical Novelty And Significance:** 2
**Empirical Novelty And Significance:** 3
**Recommendation:** 5

**Clarity, Quality, Novelty And Reproducibility:**

Clarify is fine.

Quality is OK.

Novelty is OK.

Reproducibility is good.

**Strength And Weaknesses:**

# Strength
1. The work considers a very important topic: understanding the effect of large stepsize in SGD. Despite the caveats in the theory (please see more in below), I (and I believe many others in the community) share somewhat similar understanding as the authors have proved in their theorems.
2. I find most of the presentation is good. I especially like the cartoon figures, which speak for themself in a very nice way.
3. The design in Section 5.1 is very interesting to me. I think this design could inspire more follow-up study on SGD (stepsize and noise).

# Weakness
Most of my criticism about this work will be about their theorems.

1. Thm1 could have cited [Wu, et al, 2018]. Note that the "linear stability" in [Wu, et al, 2018] can exactly explain the results in Thm 1. Because for the choice of stepsize, $x\_\\dagger$ is not linear stable but $x\_\*$ is linear stable. Granted, Thm1 discusses the behavior of the GD in an integrated manner, but the techniques are standard, using the assumptions on the one-point-convexity. I strongly suggest to add some discussion on [Wu, et al, 2018].

2.  Definition2 could be made more clear by specifying in the condition that $M$ is a fixed set (connected? compact? containing $x\_\*$? ...).

3. Theorem 1, could define what is $\\mathcal{L}$. I assume it refers to some measure over the randomness of the initialization?

4. Lemma 2 and Theorem 2. The failure probability has a crude, exponential dependence on dimension $d$ and number of iterates $T$. Such an exponential dependence makes the result rather weak and less interesting. I would like to see some discussion on how tight these are. That is, can we improve the dependence on $d$ and $T$ or not? Is the current exponential dependence purely artificial?

5. Both Theorem 1 and 2 put strong assumptions on the landscape of the loss (e.g., exactly two minimum). Moreover, proposition 3 is dedicated for a special example. I am not against to look at special cases for drawing intuition, but the theorems in their current form fail to qualify this work as a solid theory work, in my humble opinion.

6. I feel the theory and the experiments have only tangential connections. Indeed, the theorems are all for optimization and have little to do with generalization/learning. But crucial in the experiments are the test accuracy for each algorithm. Is it possible to connect the specific loss landscape required for the theorem to some nature learning settings? If not I can hardly see why these losses are important examples for understanding the learning ability of SGD.

# Missing Literature
In fact, multiple existing works have investigated the effect of large stepsize and discussed their importance for SGD (and are not from the perspective of enlarging the gradient noise). For an incomplete list, please see the following and the references therein:

[1] Wu, Jingfeng, et al. "Direction matters: On the implicit bias of stochastic gradient descent with moderate learning rate." ICLR 2021.

[2] Beugnot, Gaspard, Julien Mairal, and Alessandro Rudi. "On the Benefits of Large Learning Rates for Kernel Methods." COLT 2022.



**Summary Of The Paper:**

This work considers the effect of large stepsize for GD in non-convex optimization. It shows theoretically that, under certain conditions, with a large stepsize GD can escape/avoid undesirable minimum, and in comparison, GD/SGD with small stepsize might converge into the undesirable minimum.

Then the paper turns to present some empirical results. By re-using a mini-batch for multiple times, the work can, approximately, control the stepsize and the gradient noise in SGD in separate. By comparing this SGD variant with small/large stepsize SGD, some insights are drawn that the effect of large stepsize for SGD can go beyond simply a larger gradient noise. The second part of the experimental results are for verifying the provided theory.

**Summary Of The Review:**

Please see above. I will consider re-evaluate based on the authors reply but at this point I do not think this work qualifies being published.

---

> ### Author Response · Authors · 2022-11-10
> **Response to Reviewer hE2g (Part 1)**
>
> Dear Reviewer,
>
> Thank you for your feedback and consideration of our work. Here are some notes which we hope will clarify our contributions and answer your questions:
>
> 1. In the full-batch settings, the result of [Wu. et. al, 2018], in particular the linear stability condition, translates to the fact that GD can only converge when the maximum eigen-value of the hessian is below the threshold 2/(learning rate). This is well known in the literature from earlier works as well and our results are built on top of this fact. The main contribution of [Wu. et. al, 2018] is to show the set of minima that SGD can converge to might be further constrained than the set of minima that GD can converge to. In this work, we are focused on the changes in the trajectory of GD when using a large learning rate. As such, the results of [Wu. et. al, 2018] is on a completely different though related subject which we also cover in our literature review.
>
> 2. As we mention in the list of our contributions, the significance of Theorem 1 is the introduction of a setting where behavior similar to what is observed in practice can be observed. This for example allows future methods to be assessed theoretically in a setting which resembles practice more closely. Therefore, the contribution of Theorem 1 is indeed the integration of two minima in one landscape to demonstrate the existence of settings where GD escapes from parts of the landscape by temporarily diverging. Note that our results show that convergence actually happens. In particular, in Lemma 1 we can prove that we escape from the local minimum to a point that is closer to the global minimum. This is important since otherwise GD might be stuck in a loop of returning to the local minimum and escaping from it, never converging. Such guarantee does not exist for example in the results of [Wu. et. al, 2018]. In our proof, we also use Theorem 2 to resolve some technical difficulties which can also be applicable and useful to other works.
>
> 3. Definition 2 itself is valid for any M. However, we make further assumptions on the regions over which we use Definition 2 in our theorems.
>
> 4. In Theorem 1, L is the lebesgue measure. We will make this clear in the revision of our paper.
>
> 5. In Lemma 2, we make no assumptions on the set X. Therefore, the set X can technically be the region around the minimum. Furthermore the function itself can be for example $f(x := (x_1, \ldots, x_d)) := \frac{L}{2}\sum_{i=1}^d x_i^2$. In this case, GD converges exponentially towards X in each direction. Therefore the Lebesgue measure set of the points that converge to X within T steps increases exponentially with d (since each direction contributes with an exponential factor). This means that in the general settings the exponential dependence is not avoidable. However, further assumptions can prevent such corner cases. For example, in Theorem 2, we make the assumption about the distance of X to the minimum which improves the bound.
>
> (Reply continued in Part 2)

---

> > ### Author Response · Authors · 2022-11-10
> > **Response to Reviewer hE2g (Part 2)**
> >
> > (Continued from Part 1)
> >
> > 6. Theorem 1 is based on Lemma 1 which only makes assumptions over the local landscape around a local minima as well as the existence of a different global minimum. This means that our results can be applied when several local minima exist on the landscape. In Theorem 1, our goal is to present a setting where positive effects of large learning rates can be observed. Therefore, for simplicity we focus on the case where there is only one local minimum. The same relation holds between Theorem 2 and Lemma 2 where Lemma 2 is applicable to any arbitrary set X over any landscape. Our goal in proposition 3 is exactly to provide intuition and a concrete example to showcase how stochastic noise can fail in cases where a large learning rate succeeds. Note that this is important since as we mentioned a lot of focus in the literature has been only on implicit bias induced by the stochastic noise.
> >
> > 7. The main goal of our experiments is to establish the relevance of our theoretical results in practice. In section 5.2, we actually plot the train loss and use it to show that escaping from local minima can and does happen in the landscape of neural networks. Hence, demonstrating the relevance of this mechanism of GD to neural network training. For Section 5.1, we plot the test loss. However, we emphasize that in this work we are not considering the generalization properties of different minima. Instead, we are interested in the optimization dynamics which determine the minimum that we converge to. Since in all cases the training accuracy is almost 100%, looking at the test accuracy allows us to distinguish between different minima. We can see that when increasing the stochastic noise we do not converge to the same set of minima that we would converge to with a large learning rate. Of course, the reason we want to converge to those minima is due to their better test accuracy. However, the focus of this work is not to describe why that is the case but to instead show that a large learning rate is needed to get to those minima.
> >
> > 8. Thank you for pointing us toward the works of [Wu. et. al, 2021] and [Beugnot, et. al. 2022]. We will include them in our literature review. We emphasize that our results are different from these works. Both of these works focus on the difference of direction of convergence to the same minimum between for example GD and SGD in [Wu. et. al, 2021]. In [Beugnot, et. al. 2022], the combination of such difference and early stopping results in the improved error. However, the difference observed in practice is there even with 100% training accuracy. Combined with our observations in Section 5.2, the point of convergence changes when using a small learning rate instead of a large learning rate. Such change of the convergence point when using a large learning rate can only be observed in our results and settings.
> >
> >
> > We hope that our replies have addressed your concerns and that you would consider raising your score. We remain at your disposal to answer any other questions or comments.
> >
> > Thank you very much.

---

> ### Author Response · Authors · 2022-12-06
> **Following up**
>
> Dear Reviewer hE2g,
>
> Since the deadline for discussion is approaching, we wanted to see if our reply has addressed your concerns. Please let us know if we have answered all your questions or if you have further questions. We remain at your disposal to answer any other questions or comments.
>
> Thank you very much.

---

> > ### Comment · Reviewer_hE2g · 2022-12-07
> > **Thank you for your rebuttal**
> >
> > Hi,
> >
> > Thanks for your reply. However some of my concerns are not addressed, in particular:
> >
> > - Q4. I still have concerns on the exponential dependence on $d$. This has not been discussed in your rebuttal.
> >
> > - Q5. I still felt the setups of the theorems are too specific/limited/artificial. In the rebuttal, the authors mentioned the assumptions are only on the local landscape. However, I still did not see why the local landscape of nature ML problems should be as assumed. It would be great if the authors can give a specific and nature ML example which satisfies the assumptions.
> >
> > - Q6. Similar to Q5, I still felt the experiments and theorems are largely disconnected, because the authors did not show that neural networks satisfy the assumptions of the theorems. In my perspective, the experiments and theorems could be about two very difference objectives unless the authors could show otherwise.

---

> > > ### Author Response · Authors · 2022-12-08
> > > **Reply**
> > >
> > > Dear Reviewer hE2g,
> > >
> > > Thank you very much for your reply and the additional comments.
> > >
> > > 1- We have discussed the dependence on d in point 5 of the first part of our rebuttal. Please let us know if our explanation there alleviates your concern or if you have further questions in this regard.
> > >
> > > 2- As we mention in the list of our contributions, our goal is to “capture the distinct trajectories of large learning rate GD and small learning rate GD in theory”. Therefore, we do not claim to fully explain the training mechanics of neural networks but instead look for settings where behaviors observed in practice can be also observed. Our goal here is to introduce a setting which is closer to practice by also imitating the effects of large learning rates widely observed in practice. Such behaviors are not observable under previous assumptions such as smoothness. Therefore, future optimization algorithms can also be evaluated in settings similar to ours ensuring they also benefit from similar escaping mechanisms which seem crucial in practice. We emphasize that the fact that theoretical assumptions do not exactly match the conditions of the neural networks landscapes is true for many of the theoretical arguments in the field. In particular, many of the prior work are based on smoothness assumption over the whole landscape. Still these arguments allow for further understanding and intuitions that extend to practice and facilitate developing and verifying new algorithms. Finally, though we do not claim that our settings fully match a well known ML problem, we note that they are close to reality in some cases. For example, as mentioned in the paper and discovered in (Safran et al., 2021) one-point strong convexity only holds in most directions for over-parameterized two-layer ReLU networks.
> > >
> > > 3- As we discussed, we do not claim that our assumptions fully hold for neural networks landscape. However, through our experiments we show that we can observe a behavior consistent with our theoretical results. This strengthens the relevance and importance of our results and encourages the generalizability of future results under the introduced settings. Furthermore, one of our goals in this paper is to emphasize the important effect of large learning rate on the optimization trajectory. Our experiments complement our theoretical results in demonstrating this importance.
> > >
> > > We hope that our replies have addressed your concerns and that you would consider raising your score. We remain at your disposal to answer any other questions or comments.
> > >
> > > Thank you very much.

---

### Official Review · Reviewer_opfE · 2022-10-27

**Confidence:** 2
**Correctness:** 3
**Technical Novelty And Significance:** 2
**Empirical Novelty And Significance:** 2
**Recommendation:** 6

**Clarity, Quality, Novelty And Reproducibility:**

This paper has some good insights for explaining why one should large learning in optimizing non-convex function.
The writing seems a little messy.

**Strength And Weaknesses:**

Pros: The inituition of this paper is easy to follow and it can explain why a large learning rate is preferred in a class of non-convex functions. The proofs in this paper are solid and easy to understand.

Cons: The writing seems a little messy. Lemma 1 is not easy to read. I also doubt  whether the claim in this paper holds for training neural networks.

**Summary Of The Paper:**

This paper tries to  explain why large learning rate is important in non-convex optimiztion.
This paper shows that  GD with large step size—on
certain non-convex function classes—follows a different trajectory than GD with
a small step size, which can lead to convergence to a global minimum instead of a
local one.
The intuition behind this paper is direct and easy to follow.
The proofs in this paper are solid.

**Summary Of The Review:**

This paper tries to  explain why large learning rate is important in non-convex optimiztion.
This paper shows that  GD with large step size—on
certain non-convex function classes—follows a different trajectory than GD with
a small step size, which can lead to convergence to a global minimum instead of a
local one.
The intuition behind this paper is direct and easy to follow.
The proofs in this paper are solid.

---

> ### Author Response · Authors · 2022-11-10
> **Response to Reviewer opfE**
>
> Dear Reviewer,
>
> Thank you very much for your encouraging comments. We understand that Lemma 1 is not very easy to read but we also need to state all the assumptions to be accurate. We will try to further improve the writing of the paper, including Lemma 1, and would appreciate any other concrete comments you have in this regard.
>
> Please note that we provide experimental evidence, such as those in Section 5.2, to show that behavior such as the escaping we show in theory can also be observed for neural networks in practice. Though we make certain assumptions to obtain our theoretical results, these experiments show a similarity of behavior between our settings and the practice, demonstrating the relevance of our results to practical scenarios such as neural network training.
>
> We hope that our replies alleviates your concerns and increases your confidence in our work. We remain at your disposal for any comments or questions you might have.
>
> Thank you.

---

### Official Review · Reviewer_juk2 · 2022-10-27

**Confidence:** 4
**Correctness:** 3
**Technical Novelty And Significance:** 2
**Empirical Novelty And Significance:** 1
**Recommendation:** 3

**Clarity, Quality, Novelty And Reproducibility:**

Clarity: Good. The paper is very clear about their findings and contribution.

Quality and Novelty: Low to Medium. The authors only considered an oversimplified case of nonconvex optimization with smooth and one-point-strongly-convex loss functions. The authors did not compare their results with some formal studies on large learning rates in neural networks, such as [Li et al. NeurIPS 2019].

Reproducibility: Medium. The code is provided, but there are not multiple runs based on different random seed. In addition, the experiment design has some issues which I mentioned in the Weaknesses section.

**Strength And Weaknesses:**

Strength:

1. The paper is clearly-written and the main message is clear.
2. The literature review is sufficient.

Weaknesses:
1. The large learning rate is indeed important in neural network training, but the authors did not analyze the neural network training trajectory of gradient descent theoretically: instead, the authors simply assumed the function is smooth and one-point-strongly-convex, and there are exactly two minima. I am not sure the insight obtained in this paper can be generalized in neural networks since some arguments are made over some contrived functions (e.g., the function plotted in Figure 5).

2. I am wondering what is the real technical contribution of this paper. It seems to be that the proof technique is standard: it uses standard proof roadmaps for smooth functions and combines with the one-point-strong-convexity to obtain the relationship between two consecutive iterates.

3. The experiments are not well-designed. In Section 5.1, to disentangle the effects of learning rate and the stochastic noise, the authors should consider noiseless case instead of noisy case with different scale: the reason is that the noise does not only affect the current solution but also affects the whole trajectory of gradient descent. Why the authors turns off batch normalization?

4. The insight of Section 5.2 is very similar to Figure 1 in [Li et al. NeurIPS 2019]. Although [Li et al. NeurIPS 19] did not plot the principal component but it is clear that small and large learning rate gradient descent have different trajectories (Please refer to Section 4 in [Li et al. NeurIPS 2019]).






**Summary Of The Paper:**

This paper shows that a large learning rate can provably escape local minima and reach global minima for smooth and one-point-strongly-convex functions, while a small learning rate would not work. The paper further shows that a higher stochastic noise is not enough to close the gap with the model trained with a large learning rate.  Some experiments are conducted to disentangle the effects of stochastic noise and learning rate, to compare small and learning rates when training neural networks.

**Summary Of The Review:**

The paper considered the effects of a large learning rate and showed that under smooth and one-point-strong-convex assumptions large learning rate is better than a small learning rate. However, it is unclear that how relevant these results are because the authors did not analyze the trajectory of a certain neural network but some contrived functions (e.g., Proposition 3). The experimental design has some issues. Overall the insight is not significant compared with previous work, such as [Li et al. NeurIPS 2019], which formally analyzed the effect of the learning rate in a certain neural network.

---

> ### Author Response · Authors · 2022-11-10
> **Response to Reviewer juk2**
>
> Dear Reviewer,
>
> Thank you for your comments. Here are some notes which we hope will clarify our contributions and answer your questions:
>
> 1- As we mention in the list of our contributions, our goal is to “capture the distinct trajectories of large learning rate GD and small learning rate GD in theory”. Therefore, we do not claim to fully explain the training mechanics of neural networks but instead look for settings where behaviors observed in practice can be also observed.  Our goal here is to introduce a setting which is closer to practice by also imitating the effects of large learning rates widely observed in practice. Such behaviors are not observable under previous assumptions such as smoothness. Therefore, future optimization algorithms can also be evaluated in settings similar to ours ensuring they also benefit from similar escaping mechanisms which seem crucial in practice.
> We emphasize that the fact that theoretical assumptions do not exactly match the conditions of the neural networks landscapes is true for many of the theoretical arguments in the field. In particular, many of the prior work are based on smoothness assumption over the whole landscape. Still these arguments allow for further understanding and intuitions that extend to practice and facilitate developing and verifying new algorithms. Note that our settings are also close to reality. As mentioned in the paper and discovered in  (Safran et al., 2021) one-point strong convexity only holds in most directions for over-parameterized two-layer ReLU networks.
>
> 2- We fully disagree that there is an overlap between Figure 1 in Li et al. and our results.  Figure 1 in Li et al. shows an inferior performance when using SGD with a low learning rate. Indeed, this is the phenomenon that we are interested in understanding and we cite it in our introduction. However the figure does not explain why the difference is observed. In contrast, our results in Section 5.2, demonstrate the phenomenon of escaping from a local minima when using a large enough learning rate instead of only looking at the final loss. We provide these results as evidence to signify the importance of understanding the escaping mechanisms and their relevance on the landscape of neural networks. Without such evidence, it is for example possible that escaping from local minima might be considered irrelevant for neural networks.
>
> 3- The results of [Li et. al., 2019] are quite different from our results. Their results are based on the fact that too much noise can prevent overfitting on easy to learn features which can then allow the model to learn more generalizable patterns. The difference of large vs small learning rate comes from amplified noise in the case of large learning rate. As we discuss in the paper, we provide both empirical (Section 5.1) and theoretical results (Section 4.3) to show that the effect of large learning rate goes beyond its effect on stochastic noise.
>
> 4- The goal of the experiments in Section 5.1 is to show the relevance of the effects of large learning rate (such as escaping from local minima which we investigate in our theoretical work) even in the presence of stochastic noise. Investigating the noiseless case is therefore not useful for this goal. In particular, we show that even when stochastic noise is applied at the same level, we are not able to avoid some local minima and therefore do not converge to the same minimum without the large learning rate. Note that we do not refute the effectiveness of noise nor claim that GD can always reach the same accuracy as SGD. Rather, we state that it is important to couple it with the large learning rate.
>
> 5- The results in Section 5.1 are indeed averaged over 3 runs with different random seeds as mentioned in the paper. Figure 6 includes the averaged trajectories with the confidence interval marked with a shaded color.
>
> 6- We turn off batch normalization to avoid additional interference. For example, when repeating the same batch, the running averages might need additional tuning of parameters which we aim to avoid.
>
> We hope that our replies have addressed your concerns and that you would consider raising your score. We remain at your disposal to answer any other questions or comments.
>
> Thank you very much.

---

> > ### Author Response · Authors · 2022-12-06
> > **Following up**
> >
> > Dear Reviewer juk2,
> >
> > Since the deadline for discussion is approaching, we wanted to see if our reply has addressed your concerns. Please let us know if we have answered all your questions or if you have further questions. We remain at your disposal to answer any other questions or comments.
> >
> > Thank you very much.

---

> > > ### Comment · Reviewer_juk2 · 2022-12-12
> > > **Response**
> > >
> > > Thank you for the response.
> > >
> > > However, my main major concern remains. First, I believe this setting is far from real neural network training. Second, the technical contribution is insufficient.
> > >
> > > Therefore I decide to maintain my score.
> > >
> > > Reviewer juk2

---

> > > > ### Author Response · Authors · 2022-12-14
> > > > **Response**
> > > >
> > > > Dear Reviewer juk2,
> > > >
> > > > Regarding your first concern, we would like to point out that many theoretical work in this field use smoothness assumption to obtain various theoretical results that provide valuable insights. Our assumptions at the very least allow a more relaxed version of the landscape than complete smoothness. Therefore, as we explained in point 1 of our rebuttal, our settings allow obtaining further insights especially given that we demonstrate similar escaping behavior is observed in neural networks.
> > > >
> > > > Furthermore, even though parts of our proof (e.g. proof of Theorem 2 and its application in proof of Theorem 1) are not at all standard, the contribution of our work is not in providing complicating proof techniques. Rather, we find a setting which shows a similar behavior that was observed in practice. We additionally provide valuable insights by demonstrating the exclusive effects of large learning rates even in the stochastic setting both in theory and in practice, showing that they can not be reproduced by increasing stochasticity and establishing the importance of analyzing them.
> > > >
> > > > We hope that you will reconsider.
> > > >
> > > > Thank you.

---

### Decision · Program_Chairs · 2023-01-20

**Decision:**

Reject

**Justification For Why Not Higher Score:**

The paper's scores were clearly below the borderline for acceptance. The reviewers' scores were based on the points highlighted by me in the meta review.

**Justification For Why Not Lower Score:**

N/A

**Metareview: Summary, Strengths And Weaknesses:**

The paper considers the observation that large learning rates are useful in training of deep learning models. The paper studies this phenomenon and under certain non-convex function classes shows that the large learning rates can help escape local minima and converge to a global minima. The primary contribution of the paper according to the reviewers is to identify these function classes where this phenomenon holds. Overall the reviewers appreciated some aspects of the paper but the major concerns remained (even after a virtual discussion) --
1. The model under which the theory presented by the paper holds is still quite a bit further away from what's used in practice.
2. While point 1 might itself not be enough of a demerit, the reviewers found that the paper's technical contributions are not sufficient to be above the bar for ICLR. This was also coupled with the fact that certain theorems presented in the paper had certain not so immediately quantifiable quantities, in particular one factor whose dependence on the dimension dependence seems to be exponential in d in many cases of interest.

Overall due to these shortcomings the reviewers in the meeting generally agreed that the paper was below the bar for ICLR.

**Summary Of Ac-Reviewer Meeting:**

The meeting was conducted. The paper and its contributions were throughly addressed. The major points raised by the reviewers have been listed in the meta review and the concerns remained even after the rebuttal.